# MIND: MATH INFORMED SYNTHETIC DIALOGUES FOR PRETRAINING LLMS

**Syeda Nahida Akter**[2],[\*], **Shrimai Prabhumoye**[1],[3], **John Kamalu**[1], **Sanjeev Satheesh**[1]
**Eric Nyberg**[2], **Mostofa Patwary**[1], **Mohammad Shoeybi**[1], **Bryan Catanzaro**[1]
NVIDIA[1], Carnegie Mellon University[2], Boston University[3]
`sakter@andrew.cmu.edu, sprabhumoye@nvidia.com`

## ABSTRACT

The utility of synthetic data to enhance pretraining data quality and hence to improve downstream task accuracy has been widely explored in recent large language models (LLMs). Yet, these approaches fall inadequate in complex, multi-hop and mathematical reasoning tasks as the synthetic data typically fails to add complementary knowledge to the existing raw corpus. In this work, we propose a novel large-scale and diverse **M**ath **I**nformed sy**N**thetic **D**ialogue (MIND) generation method that improves the mathematical reasoning ability of LLMs. Specifically, using MIND, we generate synthetic conversations based on OpenWebMath (OWM), resulting in a new math corpus, MIND-OWM. Our experiments with different conversational settings reveal that incorporating knowledge gaps between dialog participants is essential for generating high-quality math data. We further identify an effective way to format and integrate synthetic and raw data during pretraining to maximize the gain in mathematical reasoning, emphasizing the need to restructure raw data rather than use it as-is. Compared to pretraining just on raw data, a model pretrained on MIND-OWM shows significant boost in mathematical reasoning (GSM8K: +13.42%, MATH: +2.30%), including superior performance in specialized knowledge (MMLU: +4.55%, MMLU-STEM: +4.28%) and general purpose reasoning tasks (GENERAL REASONING: +2.51%).

## 1 INTRODUCTION

The ability to reason is a fundamental element of human cognition, encompassing our ability to think logically, draw conclusions, and make decisions based on available information (Gendron et al., 2024). Large Language Models (LLMs) have demonstrated remarkable performance across wide range of general reasoning and specialized knowledge tasks. In particular, the improvement of LLMs in solving complex mathematical reasoning tasks (Hendrycks et al., 2021b; Cobbe et al., 2021a) has been significant in recent years (Gemini, 2024; Nvidia et al., 2024; OpenAI, 2024).

Strong mathematical reasoning ability heavily relies on the abundance of high-quality, composite, and structured pretraining corpora. An effective mathematical corpus should not only contain relevant content but also be formatted to guide models break down complex problems into smaller sub-problems and solve each part step-by-step—enhancing the model's ability to process and reason about complex problems (Wei et al., 2022). Prior studies show that structured and well-formatted corpora play a crucial role in enhancing multi-hop and logical reasoning abilities (Cobbe et al., 2021a; Li et al., 2023; Gunasekar et al., 2023), underscoring the importance of well-organized mathematical datasets in pretraining LLMs.

Curating complex, high-quality structured mathematical data is costly and resource-intensive, largely due to the uneven distribution of high-quality sources. Most advanced models (OpenAI, 2024; Gemini, 2024) are not publicly accessible, and it is unclear how their approach is enhancing math reasoning. To mitigate this challenge, synthetic data generation has emerged as a scalable, and cost-effective alternative for creating a more balanced and diverse training corpus for pretraining LLMs (Maini et al., 2024; Eldan & Li, 2023; Gunasekar et al., 2023; Shah et al., 2024). However,

---

[\*]Work done during internship at NVIDIA

while these techniques have shown promise in improving general reasoning tasks, their data often lack the step-by-step problem solving structure crucial for multi-hop reasoning and complex mathematical tasks (Maini et al., 2024), making them sub-optimal for such reasoning.

To address these challenges, we propose **MIND**, a novel approach to generate **M**ath **I**nformed sy**N**thetic **D**ialogue data at scale. In MIND, we provide a pretrained LLM with a web document and explicitly prompt it in a zero-shot manner to generate a conversation that—(a) decomposes the original context step-by-step into multi-turn conversations and (b) explores each step in depth within a single turn. As illustrated in Figure 2, MIND generates conversation from a raw text by prompting an open-source LLM on seven diverse conversational styles. The generated conversations are refined using heuristic filters and then can be used to pretrain a language model.

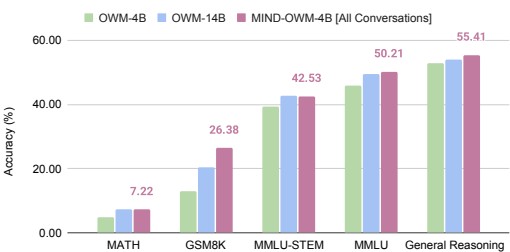

Figure 1: Continuous pretraining with all styles of conversations (MIND-OWM-4B) derived from a small subset (OWM-4B) and a 3.6× large raw corpus (OWM-14B) reveals that model trained with conversations outperforms the one trained with larger corpus in GSM8K, MMLU and general reasoning—showing the significance of high-quality structured data over quantity.

MIND demonstrates that transforming raw web text into structured conversations using an off-the-shelf open-source LLM significantly enhances the mathematical and logical reasoning abilities of LLMs compared to unstructured raw or rephrased web text. Additionally, MIND provides the flexibility to preserve the diversity of the web corpora and leverage knowledge imbalances between participants for further expansion of the corpora as they either educate each other or collaboratively bridge their shared knowledge gaps through explanation and analysis in a conversation. Moreover, MIND enables the continuous generation of synthetic data from a single document by employing infinite conversational styles, further enriching the diversity. Unlike static text rephrasing (Maini et al., 2024), conversations encourage dynamic reasoning, where participants build on each other's ideas, ask questions, and offer clarifications. This quality makes conversations particularly effective for complex reasoning tasks, as they not only preserve the original information but also expand it with new layers of understanding and explanation.

In summary, the key contributions of this work are as follows:

- We propose a novel approach, MIND, to generate structured conversational synthetic data for math reasoning. Leveraging MIND, we produce 64B tokens of synthetic data using 14B tokens from OpenWebMath corpus.

- We conduct comprehensive experiments with various conversational styles, altering participant roles to assess their impact on conversation quality and reasoning tasks. Our findings emphasize the importance of the knowledge imbalance between participants in producing high-quality mathematical data.

- We scale our approach to higher number of tokens and to two math specific datasets, demonstrating its efficacy in large and high-quality raw corpus.

- We demonstrate an effective way for integrating synthetic and raw data during pretraining to enhance mathematical reasoning ability of LLMs, emphasizing the importance of carefully reformatting raw data to optimize reasoning processes instead of using it in its original form.

In this paper, we evaluate MIND across three dimensions: (1) the effectiveness of each conversational style in mathematical reasoning, (2) whether the impact of conversation persist as data scales, and (3) whether MIND remains beneficial when the raw text originates from high-quality sources. Continuously pretraining a 7B LLM on synthetic conversations (MIND-OWM-4B), generated from a subset of OpenWebMath (OWM-4B), results in 6.29% average improvement across three mathematical reasoning benchmarks, 4.30% on specialized knowledge tasks (MMLU), and a 2.20% boost across 10 zero-shot tasks, compared to the model trained with raw OWM-4B. Additionally, our experiment with entire OpenWebMath (OWM-14B) and its corresponding synthetic conversations shows a consistent trend, indicating that the benefits of conversational data continue to hold as the data scales. In fact, with all conversations generated from OWM-4B, we can outperform model trained with OWM-14B, a 3.6× larger data—2.94% average improvement across GSM8K and MATH tasks, 1.56% across all benchmarks (Figure 1). This underlines the value of synthetic conversations,

particularly when high-quality in-domain data is limited. Moreover, our analysis with other datasets reveals that conversational data further amplifies reasoning capabilities in models even when the raw data originates from high-quality sources. We hope that MIND will pave a way to improve complex reasoning ability of smaller models with limited training data and accelerate further innovation towards building strong reasoning ability with structured high-quality data.

## 2 MIND: MATH INFORMED SYNTHETIC DIALOGUE GENERATION

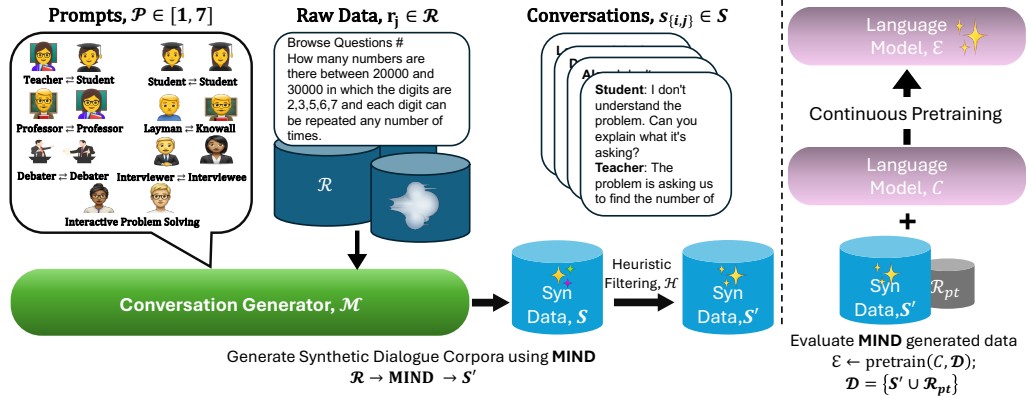

Figure 2: **Math Informed syNthetic Dialogue.** We (a) manually design prompts of various conversational styles, (b) provide the prompt along with raw context as input to LLM to obtain diverse synthetic conversations, (c) apply heuristic filtering to refine the generated data and (d) observe the downstream task performance after continuously pretraining an 7B LLM.

To generate high-quality data at scale, current synthetic data generation approach explores rephrasing texts using LLMs in varied syntax while preserving the core content (Maini et al., 2024). However, their proposed approach limits up-sampling high-quality data in a way that does not go beyond grammatical styles or surface form transformations—leading little to no improvement when it comes to performance across complex and logical reasoning tasks. We hypothesize that simple rephrasing does not leverage the full potential of the synthetic data to improve the mathematical and complex multi-hop reasoning ability of LLM. Therefore, we propose, **MIND**, a conversational synthetic data generation approach that adds semantic variations and structured complexity to the raw text which is required to improve complex reasoning ability of the LLMs. In addition, multi-turn conversations can break down the original context step-by-step while each step addresses a sub-context at a time by often injecting complimentary reasoning or explanations. This resonates with how human solves a complex problem using consecutive chain-of-thought reasoning.

As depicted in Figure 2, given a raw dataset $\mathcal{R} = \{r_1, ...r_N\}$, we define a set of conversational prompts $\mathcal{P} = \{p_1, ...p_7\}$ and utilize a pretrained LLM, denoted as $\mathcal{M}$, for synthetic data generation. We combine raw data $r_j$ with a prompt $p_i$ and pass it to $\mathcal{M}$ to produce synthetic conversation $s_{i,j}$.

$$s_{i,j} = \mathcal{M}(p_i \,||\, r_j)$$

Here, $s_{i,j}$ represents the synthetic data generated by applying prompt $p_i$ to the raw example $r_j$. For a specific prompt, the total synthetic data generated can be represented as

$$\mathcal{S} = \{s_{i,j} \mid j \in [1, N]\} \text{ for a fixed } i \in [1, 7]$$

We further apply heuristic filtering ($\mathcal{H}$) to remove bad generations:

$$\mathcal{S}' = \mathcal{H}(\mathcal{S})$$

Finally, we have a high-quality synthetic dialogue corpus $\mathcal{S}'$ which is specifically designed to improve mathematical and logical reasoning ability. To summarize MIND:

$$\mathcal{R} \rightarrow \text{MIND} \rightarrow \mathcal{S}'$$

To evaluate the effectiveness of $\mathcal{S}'$ in pretraining, we conduct continuous pretraining on a base LLM, $\mathcal{C}$, to minimize the computational costs associated with full pretraining. Our prior experiments on complete pretraining with raw data, $\mathcal{R}$ and synthetic data, $\mathcal{S}'$ validates that the ranking between models trained on $\mathcal{S}'$ or $\mathcal{R}$ remains consistent whether we use continuous pretraining or full-scale pretraining (detailed in Appendix B.1). Moreover, continuous pretraining has emerged as an effective way to improve performance of LLMs in target domains (Guo et al., 2024; Huang et al., 2023; Chen et al., 2023) and even boost their general capabilities (Ibrahim et al., 2024; Parmar et al., 2024c) with reduced training cost. Given the similar outcomes and significant savings in computational resources, we adopt continued pretraining for evaluating our approach throughout the paper.

Using $\mathcal{S}'$ and a subset of pretraining data ($\mathcal{R}_{pt}$), the model $\mathcal{C}$ is continuously pretrained, yielding an enhanced model $\mathcal{E}$, which possess improved mathematical reasoning capabilities.

$$\mathcal{E} \leftarrow \text{pretrain}(\mathcal{C}, \mathcal{D}); \text{ where } \mathcal{D} = \{\mathcal{S}' \cup \mathcal{R}_{pt}\}$$

### 2.1 COMPOSE CONVERSATIONAL PROMPTS

To generate conversation using a document $r_i$, we prompt $\mathcal{M}$ in a way that preserves all information from the original context in the conversation and remains faithful to the context. We manually compose several prompts on diverse conversation setting and topics. We finalize seven prompts ($\mathcal{P}$) featuring conversations between (1) TWO STUDENTS, (2) TEACHER STUDENT, (3) TWO PROFESSORS, (4) DEBATE, (5) PROBLEM SOLVING, (6) LAYMAN KNOWALL, and (7) INTERVIEW which can be found in Appendix A.1. These prompts are specifically designed to guide LLM in breaking down the input context step-by-step, with each step being discussed in depth through explanations and reasoning.

### 2.2 GENERATE CONVERSATION

Given an unstructured raw text ($r_j$), we instruct the LLM to convert the raw text into a multi-turn conversation ($s_{i,j}$) using a prompt ($p_i$) where $p_i \in \{\text{two\_students}, \text{teacher\_student}, ..., \text{debate}\}$.

**Seed Data Selection.** The benefit of MIND will get amplified for raw texts that require step-by-step analysis and chain of thought reasoning—the key features of a math corpus. Therefore, we choose OpenWebMath (Paster et al., 2023) as our seed corpus, $\mathcal{R}$, which contains 14.7B tokens of high quality mathematical web text.

**Large Language Model.** We use $\mathcal{M} = \text{LLAMA3-70B-INSTRUCT}$ (AI@Meta, 2024) to generate conversations from raw text, due to its superior performance across a variety of tasks compared to other open-source models. The instruction-tuned version is specifically fine-tuned and optimized for dialogue and chat-based applications.

**Generation Configuration.** We observe that with increasing context length, conversations tend to lose details from the original texts, as discussed in Appendix C.1. Therefore, for each generation, we iteratively take contexts of 500 tokens to obtain accurate and informative conversations. To evaluate the quality of the generated conversations, we test various filtering methods, from simple heuristics to LLM-based scoring. However, as noted in Appendix C.3, LLM scoring consistently rates all generations highly, making it unsuitable for our approach. Hence, we rely on heuristic filtering to discard bad generations before using them for training.

## 3 EXPERIMENTAL SETUP

**Conversation Generator Configuration.** To generate conversation, we consider zero-shot prompting $\mathcal{M}$, where we only pass a basic prompt (Appendix A.1) and the raw text. We sample conversations with `temperature=1.0` and `top_p=0.9` where the total number of input-output tokes is limited to 4096. We use the TensorRT-LLM toolkit to deploy large scale generation[1].

**Pretrained Model Architecture.** We train a standard decoder-only Transformer (Vaswani et al., 2017) architecture of 7B parameters ($\mathcal{C}$). The framework uses causal attention masks and Rotary

---

[1] https://github.com/NVIDIA/TensorRT-LLM

Position Embeddings (RoPE) (Su et al., 2021), Tiktoken tokenizer, SwiGLU (Shazeer, 2020) activations in the MLP layers, and grouped query attention (GQA) (Ainslie et al., 2023). The model consists of 32 layers, 32 attention heads, sequence length of 4096, and a hidden dimension size of 4096. It has no bias terms, has dropout rate of zero, and untied input-output embeddings. The models are trained using NVIDIA's Megatron-LM (Shoeybi et al., 2019) repository.

## 3.1 TRAINING DETAILS

**Pretraining Data.** Our pretraining data blend comprises of publicly available datasets from 13 snapshots of CommonCrawl (73.37%) (Gao et al., 2020), books/patents (9%), papers (9%), code (5.12%), stack-exchange (2.66%), and Wikipedia (0.8%). Our code data consists of 42 programming languages while the other datasets come from various sources including web documents, news articles, scientific papers, and books.

**General Pretraining.** To prepare a base model, we pretrain a 7B LLM on our pretraining data blend till 700B tokens using 512 H100 80GB SXM5 GPUs. During training, we use the AdamW optimizer (Loshchilov & Hutter, 2019) with $\beta_1 = 0.9$, $\beta_2 = 0.95$ and weight decay of 0.1. We use a 2-way tensor and pipeline parallelism to train the model. We set the maximum value of learning rate to $3e^{-4}$, minimum to $3e^{-6}$, and use a batch size of 6M tokens with a 4096 context length.

**Continued Pretraining.** After pretraining the base model ($\mathcal{C}$) on 700B tokens, we proceed with continuous pretraining using an additional 50B tokens to obtain $\mathcal{E}$. To reduce the shift between pretraining and continuous pretraining token distributions (Guo et al., 2024) we create a new data blend ($\mathcal{D}$) for this phase. To ensure the model is exposed to more math tokens, blend $\mathcal{D}$ consists of 2:1 ratio of OpenWebMath (33B tokens)—either raw ($\mathcal{R}$) or synthetic ($\mathcal{S}'$)— and 13 snapshots of CommonCrawl (17B tokens) ($\mathcal{R}_{pt}$) to maintain consistency with the pretraining blend. To ensure fair comparison, we always keep this token distribution constant in every experiment i.e., every model will see a the same amount of tokens from a data source regardless of its size. Unlike the pretraining blend, we use a high quality version of CommonCrawl data ($\mathcal{R}_{pt}$) filtered by the FineWebEdu (Penedo et al., 2024) classifier to achieve reasonable performance in generative tasks. This $\mathcal{R}_{pt}$ remains constant across all our continued pretraining experiments, while we vary the OpenWebMath with $\mathcal{R}$ or $\mathcal{S}'$ or combining both to assess their relative significance. We maintain the same training configuration as before and continue pretraining until reaching 50B tokens, using the same pretraining loss objective. In this paper, we use two versions of OpenWebMath:

- **OWM-4B:** To quickly evaluate the effectiveness of all seven prompts, we take a smaller subset of OpenWebMath containing 4B tokens. Synthetic data generated from this subset is labeled as MIND-OWM-4B throughout the paper.
- **OWM-14B**: This version contains the full 14.7B tokens of OpenWebMath and the synthetic data of this is called MIND-OWM-14B.

## 3.2 EVALUATION METRICS

To evaluate the zero-shot and few-shot learning capabilities of our models, we conduct a thorough benchmark assessment using a series of datasets using LM Eval Harness (Gao et al., 2024).

**General Purpose Reasoning Tasks.** This category comprises datasets testing broader cognitive skills and language comprehension. We consider nine standard commonsense and logical reasoning tasks in 0-shot: ARC easy (ARC-E) & challenge (ARC-C) (Clark et al., 2018), PIQA (Bisk et al., 2020), SIQA (Sap et al., 2019), HellaSwag (Zellers et al., 2019), WinoGrande (Sakaguchi et al., 2021), OpenBookQA (Mihaylov et al., 2018), TruthfulQA (Lin et al., 2022), CommonsenseQA (Talmor et al., 2019) and a reading comprehension task: RACE (Lai et al., 2017). We report the average results across ten general reasoning tasks under the metric 'GENERAL REASONING'.

**Math and Specialized Knowledge Tasks.** We consider three diverse math benchmarks to comprehensively evaluate the mathematical reasoning ability. These benchmarks encompass mathematical challenges from elementary to college level complexity demanding qualitative reasoning (8-shot GSM8K (Cobbe et al., 2021b), 4-shot MATH (Hendrycks et al., 2021c)) and conceptual science and math reasoning (5-shot MMLU-STEM (Hendrycks et al., 2021a)). In the Specialized Knowledge category, we evaluate on MMLU that spans multiple domains, from professional to academic, testing the model on specialized subjects.

## 4 EXPERIMENTS AND RESULTS

By leveraging MIND with seven conversational prompts and the raw OWM-4B, we generate a new corpus of 43 billion tokens (All Conversations). Additionally, employing the entire OWM-14B dataset and TWO STUDENTS conversation style, MIND produces an additional 21 billion tokens—resulting in a total of 64 billion tokens. This underscores MIND's potential to generate vast amount of high-quality data from relatively limited source material[2].

**Performance across Individual Prompt Style.** We observe the effect of each conversation style by generating synthetic data with seven prompts for a smaller subset of OpenWebMath, denoted as OWM-4B. To establish a baseline, we continue pretraining $\mathcal{C}$ using $\mathcal{D} = \{\mathcal{R} \cup \mathcal{R}_{pt}\}$, where $\mathcal{R} \in$ OWM-4B. To further assess the significance of MIND over other synthetic data generation approach, we add another baseline 'Rephrase' introduced by Maini et al. (2024). We generate rephrases with $\mathcal{M}$ using the highest performing prompt from their paper to maintain consistency among generation quality and training setup. We continuously train $\mathcal{C}$ with $\mathcal{D}$ where $\mathcal{R} \in$ Rephrase-OWM-4B. In subsequent experiments, we replace $\mathcal{R}$ with $\mathcal{S}'$ where $\mathcal{S}' =$ MIND-OWM-4B, corresponding to a particular conversation style, and repeat the training. To assess the utility of combining multiple conversations, we create a new dataset by selecting the longest conversation for each context from the seven generated conversations, labeling it as the LONGEST CONVERSATION dataset.

As shown in Table 1, models trained on MIND-generated data of individual styles consistently outperform those trained on rephrased or raw data across all reasoning tasks. Specifically, models trained on synthetic data exhibit significant improvements in mathematical reasoning compared to the baseline, achieving absolute gains ranging from 4.78% to 12.82% on GSM8K, 0.54% to 1.28% on MATH, and 0.79% to 4.28% on MMLU-STEM. In specialized knowledge tasks such as MMLU, synthetic data leads to improvements ranging from 1.08% to 4.55%. Furthermore, synthetic data yields an overall enhancement in general reasoning ability, with up to a 2% absolute average improvement across the ten reasoning tasks. The LONGEST CONVERSATION delivers the highest gains across all tasks, demonstrating the potential of incorporating multiple perspectives into the training corpus.

| Dataset | Style | GSM8K | MATH | MMLU-STEM | MMLU | GENERAL REASONING (Avg) | Avg-All* |
|---------|-------|-------|------|-----------|------|-------------------------|----------|
| OWM-4B | Raw | 12.96 | 4.92 | 39.39 | 45.91 | 52.90 | 29.17 |
| | Rephrase | 11.68 | 5.46 | 39.71 | 46.17 | 53.58 | 29.22 |
| MIND-OWM-4B | TEACHER STUDENT | 22.74 | 5.96 | 40.72 | 47.93 | 54.84 | 32.87 |
| | TWO STUDENTS | 21.30 | 6.20 | 41.90 | 48.77 | 54.32 | 32.65 |
| | LAYMAN KNOWALL | 17.74 | 5.46 | 41.96 | 48.87 | 54.89 | 31.74 |
| | DEBATE | 23.96 | 6.12 | 40.18 | 47.61 | 54.76 | 33.11 |
| | INTERVIEW | 20.92 | 5.86 | 40.53 | 46.99 | 54.73 | 32.12 |
| | PROBLEM SOLVING | 24.72 | 6.16 | 41.36 | 47.74 | **54.90** | 33.38 |
| | LONGEST CONVERSATION | **25.78** | **6.30** | **42.72** | **49.37** | 54.86 | **34.08** |

Table 1: **Results of 7B LLM pretrained on Diverse Conversational Styles.** Continuous training with different conversation styles improves all reasoning tasks. Selecting the longest conversation for each raw text further enhances performance in math and specialized knowledge tasks[3]. *Average of GSM8K, MATH, MMLU and General Reasoning.*

The disparity between Rephrase and MIND is closely related to the limitations of the rephrasing process. Rephrase adds linguistic variations to the older data, preserving the syntactic meaning of the document, but can not generate semantic/pragmatic variations. Moreover, rephrases are limited to the information in the raw text and unable to inject new knowledge into the data. As evidenced in our experiments, while rephrasing offers some benefits, it falls short in addressing the deeper, more complex reasoning challenges that conversational data can resolve. The structured and interactive nature of conversations facilitates a more nuanced understanding of the problem space, making it an effective approach for improving mathematical reasoning of LLMs.

**Analysis with Complete OpenWebMath.** Building on the findings from OWM-4B experiments, we establish that all seven conversational styles contribute to significant improvements compared to the raw data. This insight prompted us to explore the effect of increased data in reasoning by scaling

---

[2]To maintain consistency, we use a subset of the data (33B tokens) in all experiments.

[3]Further breakdown of individual tasks are in Appendix B.2.

our synthetic conversation generation for the complete OWM-14B corpus. To generate data, we follow the similar recipe as before and apply only one conversation style to minimize the generation cost. Among the top three highest-performing prompts across all tasks, we randomly choose TWO STUDENTS prompt style to generate conversations (MIND-OWM-14B). We then continuously train $\mathcal{C}$ on OWM-14B and MIND-OWM-14B alternatively to assess the impact at a larger data scale. In this phase, we include another experiment by continuously training $\mathcal{C}$ on 50B additional tokens using $\mathcal{D} = \{\mathcal{R}_{pt}\}$ to observe how much gain we can attain across all tasks from math-centric pretraining.

| Dataset | Style | GSM8K | MATH | MMLU-STEM | MMLU | GENERAL REASONING (Avg) | Avg-All |
|---|---|---|---|---|---|---|---|
| Pretraining Data | Raw | 9.33 | 4.74 | 37.84 | 45.41 | 53.22 | 28.17 |
| OWM-14B | | 20.47 | 7.24 | 42.82 | 49.49 | 53.95 | 32.79 |
| MIND-OWM-14B | TWO STUDENTS | **27.29** | **8.24** | **43.55** | **49.91** | **55.54** | **35.25** |

Table 2: **Results of 7B LLM trained on Complete OWM-14B and MIND-OWM-14B:** Continuous training of LLM with synthetic conversation outperforms models trained with original pretraining blend and raw OpenWebMath across all tasks.

As consistent with the previous findings, Table 2 shows that model trained on synthetic conversations is undoubtedly the best for math benchmarks while it also improves overall average for all other reasoning tasks. This underscores that, with data scaling, MIND maintains significant gains in mathematical reasoning while preserving and enhancing performance across other reasoning tasks, including commonsense, factual, and specialized knowledge.

## 5 ABLATIONS

**Does the Prompt Style matter?** From Table 1, we observe improvement across all tasks using six conversational styles. However, our experiment with TWO PROFESSORS conversations yield relatively equivalent or worse performance compared to the raw data (Table 3).

| Dataset | Style | GSM8K | MATH | MMLU-STEM | MMLU | GENERAL REASONING (Avg) | Avg-All |
|---|---|---|---|---|---|---|---|
| OWM-4B | Raw | 12.96 | 4.92 | 39.39 | 45.91 | 52.90 | 29.17 |
| MIND-OWM-4B | TWO PROFESSORS | 13.50 | 4.52 | 37.93 | 45.25 | 53.21 | 29.12 |

Table 3: **TWO PROFESSORS prompt style vs Raw data.** Continuous pretraining with TWO PROFESSORS conversations does not provide gain over raw data compared to other conversational styles.

This outcome can be attributed to the nature of the TWO PROFESSORS conversation style. Upon reviewing the generated conversations, we hypothesize that the relatively lower performance is due to the zero-knowledge gap between participants. In this setup, both participants assume that the other already has sufficient knowledge as they are the domain experts, leading to surface-level engagement and less detailed discussions.

To further investigate, we measure the BLEU and ROUGE scores between the raw text and the corresponding conversation, as shown in Figure 3, and find that the TWO PROFESSORS style exhibits the highest similarity to raw text.

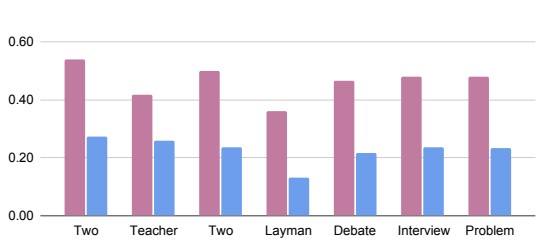

Figure 3: **Similarity between Raw Text & Synthetic Dialogues.** The TWO PROFESSORS style exhibits greater similarity to raw text, while LAYMAN KNOWALL shows the lowest similarity due to its richer context with details and explanations.

This implies that TWO PROFESSORS dialogues do not fully exploit the potential of the generation model to introduce new reasoning or breakdowns of complex problems, aligning with our qualitative observation that the professors are not engaging in deeper analysis of concepts. This contrasts with other conversational styles where there is either a clear knowledge gap between participants (LAYMAN KNOWALL, TEACHER STUDENT, INTERVIEW), forcing one to explain concepts in more depth, or both participants, being non-experts

are actively analyzing and solving the problem (PROBLEM SOLVING, DEBATE, TWO STUDENTS) which results in expanded dialogues with complementary explanations and reasoning. In the latter case, the lack of expertise creates an implicit knowledge gap—instead of one participant being more knowledgeable, both non-experts collaborate to bridge their shared knowledge gap. As depicted in Figure 3, the LAYMAN KNOWALL style, which features the greatest knowledge imbalance between participants, has the lowest BLEU and ROUGE scores. This supports our hypothesis that a larger information gap encourages the knowledgeable participant to explain concepts thoroughly, leading to more explicit and detailed conversations.

Relating these insights to our findings in Table 1, we see that incorporating explicit knowledge gaps in dialogues is beneficial for MMLU and general reasoning tasks. Conversely, collaborative problem solving, to close the implicit knowledge gap, is crucial for improving performance on math tasks. This highlights a key characteristic of high-quality math data—merely breaking down the problem is insufficient for effective math reasoning. Instead, dynamic knowledge exchange and analysis within the dialogues are essential to achieve maximum improvement in math reasoning.

**Does Conversation benefit other datasets?** OpenWebMath used in our current experiments is predominantly collected from mathematical web pages that can contain noisy web contexts. Generating synthetic conversations for such noisy contexts upsamples high-quality data and hence we observe a huge gain in performance with high-quality conversations. Here, we investigate if MIND works on high-quality datasets such as books or papers. We consider a new seed corpus, MATHPILE (Wang et al., 2023), that consists of 9.3B tokens extracted from high-quality data sources such as ArXiv papers, textbooks, StackExchange, Wikipedia, ProofWiki, and CommonCrawl pages.

| Dataset | Style | GSM8K | MATH | MMLU-STEM | MMLU | GENERAL REASONING (Avg) | Avg-All |
|---|---|---|---|---|---|---|---|
| Pretraining Data | Raw | 9.33 | 4.74 | 37.84 | 45.41 | 53.22 | 28.17 |
| MATHPILE | | 8.79 | 4.96 | 42.82 | 49.49 | 54.16 | 29.35 |
| MIND-MATHPILE | TWO STUDENTS | 12.74 | 5.74 | 43.55 | 49.91 | 53.98 | **30.59** |

Table 4: **MATHPILE vs Synthetic Conversation from MATHPILE (MIND-MATHPILE).** Conversation generated from high-quality raw data further improves the performance of math tasks.

By employing $\mathcal{M}$, we generate conversations from raw text with the TWO STUDENTS prompt. Later, we replicate the experiments by replacing OWM with MATHPILE and MIND-MATHPILE accordingly. Table 4 shows that MIND-MATHPILE outperforms the raw counterpart in all three math benchmarks along with specialized knowledge tasks, achieving comparable scores in general reasoning task. In addition, majority of MATHPILE data is from ArXiV papers and recent work has found this source ineffective in improving mathematical reasoning (Shao et al., 2024). We observe a similar trend, where non-math focused pretraining corpora yields better GSM8K score than raw MATHPILE corpus. However, our synthetic conversation on MATHPILE rather amplifies the quality of the corpus resulting in 3.95% absolute improvement on GSM8K in comparison with raw data. This highlights the superior structured complexity of conversations, which proves particularly effective for multi-hop and mathematical reasoning tasks, over high-quality data from Arxiv papers.

**Is replacing with Synthetic Data the best option?** Our findings in Table 1, 2 indicate that completely replacing OpenWebMath with synthetic data yields the best performance across benchmarks. However, Maini et al. (2024) emphasizes the importance of combining real data and synthetic rephrases to achieve consistent improvements across a broader range of tasks—a similar trend we observe in our experiment with rephrased data, as shown in Table 5. To investigate this further, we conduct experiments with four data combinations using OWM-4B while the $\mathcal{R}_{pt}$ remains constant:

- **OWM-4B + MIND-OWM-4B [1:1].** We combine $\mathcal{R}$ and $\mathcal{S}'$ in a 1:1 ratio, ensuring an equal number of tokens to be seen during pretraining from both sources. For the synthetic data, we utilize the LONGEST CONVERSATION, as this shows the most improvement across tasks (Table 1).
- **OWM-4B + MIND-OWM-4B [Concat].** We concatenate each raw context with all seven synthetic conversations sequentially.
- **MIND-OWM-4B [Longest Conversation].** From the seven conversations generated for each context, we select the longest conversation in token count.

- **MIND-OWM-4B [All Conversations].** This data incorporates all conversation across all styles.

| Dataset | GSM8K | MATH | MMLU-STEM | MMLU | GENERAL REASONING (Avg) | Avg-All |
|---|---|---|---|---|---|---|
| OWM-4B | 12.96 | 4.92 | 39.39 | 45.91 | 52.90 | 29.17 |
| OWM-14B | 20.47 | 7.24 | 42.82 | 49.49 | 53.95 | 32.79 |
| Rephrase-OWM-4B | 11.68 | 5.46 | 39.71 | 46.17 | 53.58 | 29.22 |
| OWM-4B+Rephrase-OWM-4B [1:1] | 14.25 | 6.20 | 42.31 | 48.74 | 53.68 | 30.72 |
| OWM-4B+MIND-OWM-4B [1:1] | 21.68 | 6.14 | 42.56 | 49.57 | 54.50 | 32.97 |
| OWM-4B+MIND-OWM-4B [Concat] | 24.49 | 6.22 | **43.67** | **50.46** | 55.10 | 34.07 |
| MIND-OWM-4B [Longest Conversation] | 25.78 | 6.30 | 42.72 | 49.37 | 54.86 | 34.08 |
| MIND-OWM-4B [All Conversations] | **26.38** | **7.22** | 42.53 | 50.21 | **55.41** | **34.80** |

Table 5: **Comparison of 7B LLM trained with raw and combination of synthetic data.** Synthetic conversation outperforms raw data in all combinations. Specifically, combinations of all conversations generated from OWM-4B surpasses the performance of OWM-14B ($3.6\times$ larger corpus) across all tasks, underscoring the superior quality and diversity of the conversations.

Our finding in Table 5 indicates that all combinations provide substantial boost in performance across all tasks. However, for math-centric benchmarks (GSM8K and MATH), training solely with synthetic conversations elicits the best improvements. This is likely as these tasks require complex and multi-step reasoning and conversations are designed to replicate these type of thinking. In parallel, having both raw data and conversation is beneficial for specialized and general purpose reasoning tasks, aligning with the findings in Maini et al. (2024). Since synthetic data tends to remove special tags, styles, and code indentations, the inclusion of raw data helps improve the generalizability of LLMs across diverse domains. Additionally, to measure the maximum gain we can achieve from conversations for a limited data, we continuously train $\mathcal{C}$ with all synthetic dialogues generated from OWM-4B. As shown in Table 5, using conversations generated from OWM-4B, we can outperform the model trained with $3.6\times$ bigger corpus (OWM-14B) on GSM8K, MMLU and general reasoning tasks while showing comparable performance on other tasks. Inspired by this, we further compare MIND with DEEPSEEKMATH (Shao et al., 2024) that extract 120B unique math tokens from CommonCrawl (Appendix C.4). The results from Table 14 demonstrate that diverse conversations from MIND based on a small seed corpus can yield comparable math accuracy to the DEEPSEEKMATH model. This illustrates the potential to enhance reasoning with limited data by generating synthetic conversations of infinite styles.

**Does the improvement persist with smaller $\mathcal{M}$?** In the previous experiments, we used a constant $\mathcal{M}$, a powerful instruction-tuned model. However, it remains unclear whether the improvements in downstream reasoning tasks stem from the quality of the generated dialogues or are primarily due to model distillation from the powerful LLM. To asses the impact of $\mathcal{M}$ on the downstream task performance, we re-run MIND with a smaller $\mathcal{M}$=LLAMA3-8B-INSTRUCT on PROBLEM SOLVING style, the best performing style in Table 1 and continuously pretrained a 7B LLM following the training setup in Section 3.1.

| Dataset | $\mathcal{M}$ | GSM8K | MATH | MMLU-STEM | MMLU | GENERAL REASONING (Avg) | Avg-All |
|---|---|---|---|---|---|---|---|
| OWM-4B | - | 12.96 | 4.92 | 39.39 | 45.91 | 52.90 | 29.17 |
| MIND-OWM-4B | LLAMA3-8B-INSTRUCT | 22.37 | 5.72 | 41.36 | 48.48 | 55.21 | 32.95 |
| | LLAMA3-70B-INSTRUCT | 24.72 | 6.16 | 41.36 | 47.74 | 54.90 | 33.38 |

Table 6: **Results of 7B LLM trained on MIND-OWM-4B using $\mathcal{M}$ of different sizes:** Regardless of the sizes of $\mathcal{M}$, model trained on MIND-OWM-4B outperforms the one trained with raw data.

As shown in Table 6, even with a smaller $\mathcal{M}$, the MIND-generated data provides a significant boost in math and general reasoning abilities compared to the raw/rephrased data. This demonstrates that the gains are not solely dependent on the capabilities of the larger $\mathcal{M}$ but are largely driven by the quality and structure of the MIND-generated dialogues. Additionally, regardless of model size and method of synthetic data generation, all LLM-generated synthetic data involves some form of knowledge distillation. However, we demonstrate an effective distillation approach that significantly enhances the reasoning ability of LLMs compared to existing approaches (Maini et al., 2024).

## 6    RELATED WORKS

**Mathematical Data Curation.**  Selecting high quality data for pretraining LLMs is essential for producing state-of-the-art large language models (Brown et al., 2020; Chowdhery et al., 2023; Parmar et al., 2024a;b; Rae et al., 2021; Feng et al., 2024). Several mathematical datasets have been introduced in recent years (Paster et al., 2023; Wang et al., 2023; Azerbayev et al., 2023a; Welleck et al., 2021) which have been carefully collected from the web using different heuristics. OpenWebMath contains 14.7B tokens of mathematical web pages filtered from CommonCrawl based on math strings, LATEXcontents and a math document classifier. Building on this corpus, DEEPSEEKMATH (Shao et al., 2024) trains a fastText (Joulin, 2016) classifier to further extract mathematical documents from CommonCrawl. They cluster the extracted documents based on the URL domain and label a domain math-related where over 10% of the web pages have been collected are classified as math content. Finally, web pages linked to these URLs, yet uncollected, will be added to the seed corpus which will be used to retrain the fastText classifier to fetch diverse math contexts. MATH-PILE (Wang et al., 2023), a multi-source corpus (8.9B tokens), has been aggregated from textbooks, Wikipedia, ProofWiki, CommonCrawl, StackExchange, and arXiv, with the majority (over 85%) sourced from high quality data source arXiv. Although these datasets can effectively capture the diverse mathematical information from web, it is difficult to detect and filter out noisy dataset. Recently, many powerful models (OpenAI, 2024; Jiang et al., 2023; Gemini, 2024; Anthropic, 2024; Team, 2024b), in addition to not open sourcing their data, are also refraining from disclosing detailed information about their corpus. For the open-source community, constructing high-quality and diverse pretraining corpora is a crucial factor in bridging the performance gap with closed-source models which is the main objective of our work.

**Synthetic Math Data.**  Generating synthetic math data using LLMs has been widely explored in recent days (Trinh et al., 2024; Li et al., 2024; Gunasekar et al., 2023; Madaan et al., 2024; Patel et al., 2024; Toshniwal et al., 2024) specifically during alignment using supervised fine-tuning (SFT) (Taori et al., 2023). Some of the latest approaches focus on generating data from seed problems. For instance, Yu et al. (2023) rewrites existing benchmark questions from multiple perspectives using LLMs to create new mathematical problems, while Huang et al. (2024); Shah et al. (2024) leverage GPT-4 to extract topics and key points from seed samples and recombine them into new questions. To further improve diversity, Chan et al. (2024) uses GPT-4 to generate questions and answers at scale, incorporating over one million personas. Previous approaches to generate synthetic data is primarily designed for fine-tuning rather than pretraining, distinguishing it from our effort. Similar to ours, Dai et al. (2022) converts documents into dialogues by predicting unobserved questions without altering the original document. However, MIND expands knowledge by adding complementary reasoning and explanations, leveraging diverse conversational styles to enhance reasoning and enrich diversity, which is infeasible with Dai et al. (2022). In the context of pretraining, recent works have generated synthetic datasets (Gunasekar et al., 2023; Li et al., 2023) to train smaller language models that demonstrate equivalent performance as the larger models on certain mathematical benchmarks. However, these methods remain largely opaque, costly, and reliant on proprietary models to produce billions of tokens. Additionally, such data generation can be biased towards specifically generating data related to tasks that we want to perform well on. In contrast, MIND provides a feasible alternative to upsample high quality structured data from diverse web contexts, that embeds multi-step and chain-of-thought reasoning, using an off-the-shelf open source LLM.

## 7    CONCLUSION

In this paper, we focus on improving the mathematical reasoning abilities of open-source LLMs. We propose a simple approach to generate complex and structured data at scale, called MIND, that produces a new conversational synthetic math corpus, MIND-OWM, using an off-the-shelf open-source LLM. Models trained on MIND-OWM, a corpus generated through our approach, consistently outperform those trained on raw data, achieving up to a 6.29% improvement across mathematical reasoning benchmarks and outperforming models trained on 3.6× larger datasets. Importantly, these gains persist across general-purpose reasoning tasks and when scaling up the data, highlighting the versatility of synthetic conversations. This work demonstrates the potential of structured conversational data to enhance reasoning, especially in cases where domain-specific high-quality data is limited, paving the way for more effective and resource-efficient pretraining of LLMs.

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

## A  PROMPTS AND DATASETS

### A.1  PROMPTS FOR CONVERSATION

**TWO PROFESSORS**

> Convert the context above as a multi-turn discussions between two professors. Make sure that their discussions strictly adhere to the context above and remains faithful to information in the context. Please DONOT add any new information/reference other than the context.

**TEACHER STUDENT**

> Convert the context above as a multi-turn discussions between a teacher and a student. The student has questions about the context and the teacher solves each of them step-by-step. Make sure that their discussions strictly adhere to the context above and remains faithful to information in the context. Please DONOT add any new information/reference other than the context.

**TWO STUDENTS**

> Convert the context above as a multi-turn discussions between two students who are working on their assignment related to the given context. Make sure that their discussions strictly adhere to the context above and remains faithful to information in the context. Please DONOT add any new information/reference other than the context.

**INTERVIEW**

> Conduct an interview-style conversation where one participant acts as the interviewer, asking questions exclusively related to the content provided, while the other participant serves as the subject matter expert, providing detailed responses based on the content. Make sure that their discussions strictly adhere to the context above and remains faithful to information in the context. Please DONOT add any new information/reference other than the context.

**PROBLEM SOLVING**

> Convert the context above as a multi-turn problem-solving conversation where participants analyze challenges or scenarios presented in the content and brainstorm solutions within the context of the provided material, avoiding speculation or unrelated discussions. Make sure that their conversation strictly adhere to the context above and remains faithful to information in the context. Please DONOT add any new information/reference other than the context.

### LAYMAN KNOW-ALL

> Imagine you are presenting the content above step-by-step to a layman. While you are presenting, the layman has a lot of followup questions regarding your presentation. You answer the questions step-by-step with chain-of-thoughts. Design this interaction between you and the layman as a multi-turn conversational manner. Make sure that the interaction strictly adhere to the context above and remains faithful to information in the context. Please DONOT add any new information/reference other than the context.

### DEBATE

> Convert the context above as a multi-turn debate-style conversation where the participants present arguments and counterarguments based solely on the content provided, without introducing external information or personal opinions. Each participant defends others arguments step-by-step with chain-of-thoughts. Make sure that the conversation strictly adhere to the context above and remains faithful to information in the context. Please DONOT add any new information/reference other than the context.

## A.2 EVALUATION METRIC DETAILS

We evaluate the LLM trained on raw and synthetic data using ten diverse general reasoning tasks, three mathematical tasks and one specialized knowledge tasks.

**General Purpose Reasoning Tasks.** All the benchmarks under this category are evaluated in zero-shot manner.

- **ARC Easy (ARC-E) and ARC Challenge (ARC-C)** (Clark et al., 2018): This dataset is proposed by the AI2 Reasoning Challenge (ARC). There are two sets of this data: (1) ARC-E and (2) ARC-C, containing science exam questions from grades 3 to 9. The ARC Challenge set includes more difficult questions compared to ARC-E that necessitate higher-order reasoning.

- **RACE** (Lai et al., 2017): This dataset has been collected from English reading comprehension exams designed for middle and high school Chinese students.

- **PIQA** (Bisk et al., 2020): Physical Interaction Question Answering evaluates physical commonsense reasoning ability of the language model.

- **Winogrande** [Wino.](Sakaguchi et al., 2019): This benchmark is structured as a fill-in-the-blank task with binary options, requiring the LLM to select the correct option for a given sentence, primarily focusing on commonsense reasoning and pronoun disambiguation tasks.

- **HellaSwag** (Zellers et al., 2019): This dataset evaluates a model's ability to resolve scenarios in a way that is both contextually appropriate and logically consistent, testing its grasp of language comprehension and commonsense reasoning.

- **OpenBookQA** [OBQA](Mihaylov et al., 2018): This dataset is designed to evaluate deeper understanding of elementary science facts by requiring models to apply these facts to novel situations using both open book knowledge and external commonsense reasoning.

- **TruthfulQA** [TFQA] (Lin et al., 2022): Evaluates models' ability to generate factually correct answers by presenting 817 questions across 38 categories, designed to challenge common misconceptions.

- **CommonSenseQA** [CSQA] (Talmor et al., 2019): This dataset has been designed to test commonsense reasoning through multiple-choice questions created from CONCEPTNET (Speer et al., 2017) relations, which requires prior knowledge beyond contextual associations for accurate answering.

- **Social-IQA** [SIQA] (Sap et al., 2019): Evaluates LLM's ability to reason about people's actions and their social implications.

**Math and Specialized Knowledge Tasks.** For these tasks, we evaluate the LLM in few-shot manner.

- **GSM8K** (Cobbe et al., 2021a): This benchmark comprises of high quality linguistically diverse grade school math word problems that evaluates the multi-step and logical reasoning ability of LLM. In this setup, we prompt the LLM with eight chain-of-thought examples from Wei et al. (2022) and take the majority vote of the answers from greedy decoding following the approach in Wang et al. (2022).

- **MATH** (Hendrycks et al., 2021c): This dataset contains challenging competition mathematics problems that requires step-by-step processing of the problem to derive the solution. We choose 4-shot prompt from Lewkowycz et al. (2022b) for our evaluation process.

- **MMLU** (Hendrycks et al., 2021a): This task is designed to evaluate a LLM's multitask accuracy across 57 diverse subjects, including elementary mathematics, US history, and law in multiple-choice question format, requiring extensive world knowledge and problem-solving skills for high performance. We explicitly consider MMLU-STEM as it contains comprehensive math and science problems that requires multi-hop and complex reasoning ability. Using the evaluation pipeline of LM Eval Harness, we evaluate the LLM with 5-shot prompts for this task.

# B   ADDITIONAL EXPERIMENTS AND RESULTS

## B.1   RESULTS OF PRETRAINING LLM FROM SCRATCH

We pretrain a 8B LLM from scratch with 300B tokens using (i) 4 snapshots of CommonCrawl (ii) OWM-4B and (iii) wikipedia, books and epubs corpus corresponding to 486B, 4B and 84B original tokens respectively. To emphasize math over other datasets, we provide 8 epochs of OWM-4B in the pretraining blend resulting in 35B OWM tokens that will be seen by the LLM during pretraining. For all other datasets, we maintain 0.46 epochs. For our experimentation with synthetic corpus, we analyze four variations in the OWM corpus while keeping the other data constant:

- **MIND-OWM-4B [TWO STUDENTS ].** This data includes conversations between two students.

- **OWM-4B + MIND-OWM-4B [1:1].** We sample raw and synthetic conversations in a 1:1 ratio, ensuring an equal number of tokens to be seen during pretraining from both sources. For the synthetic data, we utilize the TWO STUDENTS conversations.

- **OWM-4B + MIND-OWM-4B [Concat].** We concatenate each raw context with all seven synthetic conversations sequentially.

- **MIND-OWM-4B [Longest Conversation].** From the seven conversations generated for each context, we select the longest conversation in token count.

| Dataset | ARC-E | Race | PIQA | Wino. | HellaSwag | ARC-C | OBQA | TFQA | CSQA | SIQA | Avg-All |
|---------|-------|------|------|-------|-----------|-------|------|------|------|------|---------|
| OWM-4B | 66.79 | 35.98 | 77.69 | 62.19 | 68.23 | 38.91 | 37.20 | 35.92 | 19.57 | 44.42 | 48.69 |
| MIND-OWM-4B [TWO STUDENTS ] | 68.14 | 36.75 | 77.86 | 63.06 | 69.11 | 40.19 | 39.40 | 37.80 | 19.66 | 45.55 | 49.75 |
| OWM-4B+MIND-OWM-4B [1:1] | 69.74 | 37.32 | 77.64 | 63.69 | 69.51 | 40.87 | 38.20 | 34.97 | 20.39 | 44.47 | 49.68 |
| OWM-4B+MIND-OWM-4B [Concat] | 69.28 | 38.37 | 78.02 | 64.09 | 68.66 | 39.76 | 39.00 | 38.38 | 22.52 | 44.63 | 50.27 |
| MIND-OWM-4B [Longest Conversation] | 68.39 | 36.75 | 77.64 | 62.04 | 68.91 | 40.02 | 39.40 | 38.23 | 20.23 | 44.52 | 49.61 |

Table 7: **Evaluation of 8B LLM on General Reasoning tasks:** Conversations provide improvement over raw data in general purpose reasoning tasks including commonsense, factual and social reasoning tasks.

As shown in Table 7, conversational synthetic data improves general purpose reasoning ability of LLM. Specifically, the concatenation of raw text and conversations yields the best average score for all combinations—highlighting the efficacy of both data towards generalizability of LLM across wide range of reasoning tasks.

| Dataset | GSM8K | MATH | MMLU-STEM | MMLU-Humanities | MMLU-Social-Sciences | MMLU-Others | MMLU | Avg-All |
|---|---|---|---|---|---|---|---|---|
| OWM-4B | 4.78 | 4.92 | 26.29 | 25.93 | 26.75 | 27.16 | 26.46 | 12.05 |
| MIND-OWM-4B [TWO STUDENTS ] | 10.77 | 5.30 | 26.93 | 26.78 | 26.81 | 27.87 | 27.06 | 14.38 |
| OWM-4B+MIND-OWM-4B [1:1] | 8.49 | 5.02 | 28.01 | 28.44 | 28.40 | 28.39 | 28.32 | 13.94 |
| OWM-4B+MIND-OWM-4B [Concat] | 8.04 | 4.98 | 29.18 | 29.22 | 29.51 | 31.54 | 29.79 | 14.27 |
| MIND-OWM-4B [Longest Conversation] | 8.57 | 4.60 | 26.77 | 27.16 | 29.12 | 29.29 | 27.97 | 13.71 |

Table 8: **Evaluation of 8B LLM on Math and Specialized Knowledge tasks:** Conversations are always beneficial for mathematical reasoning where specialized knowledge tasks further benefit from having both raw and synthetic data in the corpus.

In addition, for mathematical benchmarks, only synthetic data produce the best imrpovement over the raw data (Table 8). The nature of conversational data being composite and structured helps the LLM to perform well in tasks that requires step-by-step processing of a complex problem. Conversely, specialized knowledge tasks require both raw and synthetic data to attain the maximum gain.

## B.2   BREAKDOWN OF INDIVIDUAL TASKS RESULTS OF CONTINUED PRETRAINED LLM

In this section, we further breakdown the performance of models trained on individual and combinations of conversation styles across general purpose reasoning tasks and specialized knowledge tasks.

**Performance across Individual Prompt Style.**   As shown in Table 9, synthetic data overall achieves highest gain for general purpose reasoning task compared to using raw or rephrased data. Table 10 further validates the efficacy of synthetic conversations on mathematical reasoning tasks where model trained with all styles of conversational data generated from OWM-4B gets the highest gain across all other models—highlighting the potential of upsampling high-quality data by generating synthetic data of diverse styles using a small seed data.

| Dataset | Style | ARC-E | Race | PIQA | Wino. | HellaSwag | ARC-C | OBQA | TFQA | CSQA | SIQA | Avg-All |
|---|---|---|---|---|---|---|---|---|---|---|---|---|
| OWM-4B | Raw | 71.89 | 37.89 | 78.24 | 65.98 | 71.42 | 46.33 | 41.40 | 36.96 | 32.35 | 46.57 | 52.90 |
| Rephrase-OWM-4B | Rephrase | 72.05 | 38.28 | 78.07 | 63.14 | 71.16 | 45.31 | 42.20 | 47.09 | 33.33 | 45.19 | 53.58 |
| MIND-OWM-4B | TWO PROFESSORS | 72.18 | 38.85 | 77.20 | 66.38 | 71.54 | 44.20 | 40.40 | 42.51 | 32.35 | 46.47 | 53.21 |
| | TEACHER STUDENT | 75.17 | 38.76 | 78.35 | 66.46 | 72.08 | 47.70 | 40.20 | 44.88 | 38.74 | 46.06 | 54.84 |
| | TWO STUDENTS | 72.90 | 38.56 | 78.24 | 65.82 | 72.24 | 46.67 | 41.00 | 44.10 | 38.25 | 45.45 | 54.32 |
| | LAYMAN KNOWALL | 74.12 | 39.04 | 78.45 | 65.27 | 72.19 | 46.42 | 41.00 | 46.25 | 41.28 | 44.88 | 54.89 |
| | DEBATE | 74.92 | 38.37 | 78.45 | 65.75 | 71.89 | 47.78 | 40.40 | 45.47 | 38.41 | 46.16 | 54.76 |
| | INTERVIEW | 73.82 | 37.99 | 78.13 | 65.11 | 72.18 | 48.72 | 42.00 | 47.81 | 36.04 | 45.45 | 54.73 |
| | PROBLEM SOLVING | 74.41 | 38.37 | 78.07 | 65.59 | 71.67 | 49.40 | 41.20 | 47.04 | 37.02 | 46.26 | 54.90 |
| | LONGEST CONVERSATION | 74.71 | 37.99 | 78.18 | 64.80 | 72.10 | 47.61 | 41.40 | 45.49 | 39.80 | 46.52 | 54.86 |
| | ALL CONVERSATIONS | 75.17 | 39.04 | 77.86 | 65.43 | 72.31 | 49.40 | 41.00 | 46.68 | 40.79 | 46.42 | 55.41 |
| OWM-4B+MIND-OWM-4B [1:1] | Combination | 74.12 | 37.99 | 78.18 | 66.54 | 72.28 | 48.12 | 41.40 | 39.27 | 40.70 | 46.37 | 54.50 |
| OWM-4B+MIND-OWM-4B [Concat] | | 74.92 | 38.28 | 77.58 | 67.32 | 72.63 | 48.55 | 41.80 | 42.26 | 40.95 | 46.72 | 55.10 |

Table 9: **Results of 7B LLM on General Reasoning Tasks:** We evaluate both the baseline and model trained with synthetic data across diverse tasks that focus on general reasoning, language understanding and commonsense.

**Analysis with Complete OpenWebMath.**   Our experiment with complete OWM-14B shows the similar trend as before. The comprehensive nature of this larger dataset continues to reinforce the advantages of synthetic data, as models trained on it also exhibit enhanced performance across both general purpose reasoning (Table 11) and mathematical reasoning tasks (Table 11). This consistency across different dataset sizes highlights the robustness of the benefits gained from incorporating diverse conversational styles, further supporting the notion that expanding training data through synthetic means can lead to significant advancements in the capabilities of language models.

| Dataset | Style | GSM8K | MATH | MMLU-STEM | MMLU-Humanities | MMLU-Social-Sciences | MMLU-Others | MMLU | Avg-All |
|---|---|---|---|---|---|---|---|---|---|
| OWM-4B | Raw | 12.96 | 4.92 | 39.39 | 41.15 | 52.84 | 52.85 | 45.91 | 21.26 |
| Rephrase-OWM-4B | Rephrase | 11.68 | 5.46 | 39.71 | 40.77 | 54.76 | 52.40 | 46.17 | 21.10 |
| MIND-OWM-4B | TWO PROFESSORS | 13.50 | 4.52 | 37.93 | 41.89 | 52.32 | 50.76 | 45.25 | 21.09 |
| | TEACHER STUDENT | 22.74 | 5.96 | 40.72 | 42.21 | 56.78 | 55.13 | 47.93 | 25.54 |
| | TWO STUDENTS | 21.30 | 6.20 | 41.90 | 43.40 | 57.07 | 55.65 | 48.77 | 25.42 |
| | LAYMAN KNOWALL | 17.74 | 5.46 | 41.96 | 44.27 | 56.19 | 55.62 | 48.87 | 24.02 |
| | DEBATE | 23.96 | 6.12 | 40.18 | 42.40 | 55.38 | 55.33 | 47.61 | 25.90 |
| | INTERVIEW | 20.92 | 5.86 | 40.53 | 41.21 | 55.48 | 53.91 | 46.99 | 24.59 |
| | PROBLEM SOLVING | 24.72 | 6.16 | 41.36 | 42.21 | 55.18 | 55.23 | 47.74 | 26.21 |
| | LONGEST CONVERSATION | 25.78 | 6.30 | 42.72 | 43.53 | 57.52 | 56.90 | 49.37 | 27.15 |
| | ALL CONVERSATIONS | 26.38 | 7.22 | 42.53 | 44.38 | 58.63 | 58.51 | 50.21 | 27.94 |
| OWM-4B+MIND-OWM-4B [1:1] | Combination | 21.68 | 6.14 | 42.56 | 43.85 | 57.59 | 57.42 | 49.57 | 25.80 |
| OWM-4B+MIND-OWM-4B [Concat] | | 24.49 | 6.22 | 43.67 | 44.87 | 59.21 | 57.16 | 50.46 | 27.06 |

Table 10: **Results of 7B LLM on Specialized Knowledge Tasks:** In this setup, we assess the domain specific knowledge of LLM specifically on mathematics, science and general knowledge. We emphasize on the GSM8K, MATH and MMLU-STEM task, as these tasks predominantly checks the mathematical reasoning ability of the LLM.

| Dataset | ARC-E | Race | PIQA | Wino. | HellaSwag | ARC-C | OBQA | TFQA | CSQA | SIQA | Avg-All |
|---|---|---|---|---|---|---|---|---|---|---|---|
| Pretraining Data | 70.88 | 38.76 | 78.78 | 67.80 | 73.90 | 43.86 | 42.60 | 41.35 | 29.65 | 44.63 | 53.22 |
| OWM-14B | 73.40 | 37.32 | 77.91 | 65.90 | 72.15 | 47.10 | 41.40 | 38.39 | 39.64 | 46.26 | 53.95 |
| MIND-OWM-14B | 75.84 | 39.52 | 78.56 | 65.67 | 72.38 | 48.55 | 42.80 | 45.06 | 39.89 | 47.08 | 55.54 |

Table 11: **Evaluations on General Reasoning Tasks with complete OWM-14B:** Conversational data is beneficial for general purpose reasoning tasks.

| Dataset | GSM8K | MATH | MMLU-STEM | MMLU-Humanities | MMLU-Social-Sciences | MMLU-Others | MMLU | Avg-All |
|---|---|---|---|---|---|---|---|---|
| Pretraining Data | 9.33 | 4.74 | 37.93 | 41.23 | 51.80 | 53.07 | 45.43 | 34.79 |
| OWM-14B | 20.47 | 7.24 | 42.82 | 44.48 | 56.61 | 56.78 | 49.49 | 39.70 |
| MIND-OWM-14B | 27.29 | 8.24 | 43.55 | 43.95 | 57.95 | 57.45 | 49.91 | 41.19 |

Table 12: **Evaluations on Math and Specialized Knowledge Tasks with complete OWM-14B:** Conversations improve mathematical reasoning over raw data.

# C ADDITIONAL ABLATIONS

## C.1 CONTEXT LENGTH VS CONVERSATION QUALITY

To generate conversations, we utilize $\mathcal{M}$, which supports input sequences of up to 8K tokens. However, the OpenWebMath corpus, composed of mathematical web pages from Common Crawl, often contains documents exceeding this 8K token limit, leading to errors when processing them with the LLM. A straightforward approach is to split these inputs into 8K-token windows, but initial experiments with this method reveal significant drawbacks. Conversations generated from the 8K-token inputs tend to summarize the lengthy context, resulting in a loss of substantial information from the original text.

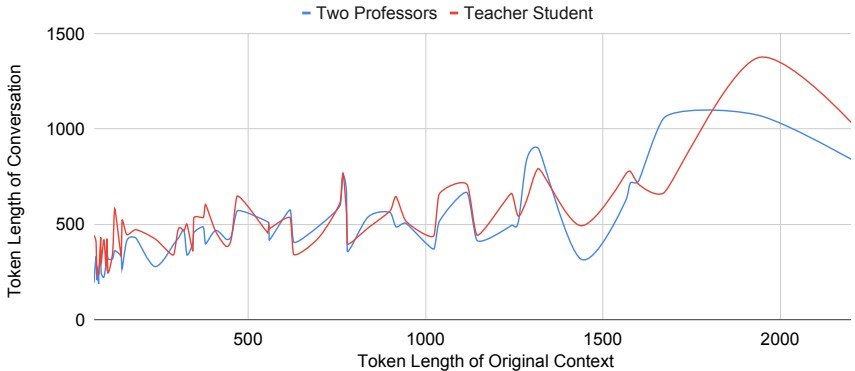

Figure 4: With increasing context length the generated conversation length decreases!

Therefore, we conduct an experiment on 140k samples from the OpenWebMath corpus of varying input length to determine the optimal input token length that generates conversations of following characteristics: (1) retains all relevant information from the original context, (2) remains grounded to the source material and (3) enhances the conversation with complementary explanations and reasoning. For each sample, we generate conversations using two prompt (TWO PROFESSORS and TEACHER STUDENT) and observe the token length of the generations. As depicted in Figure 4, with increasing input token length (X-axis), the token length of the generated conversation (Y-axis) does not scale up linearly. For example, an input context of 2K tokens results in a conversation that has 1K tokens resulting in a lot of information loss during conversion. Analyzing the Figure 4, we see that the input token length of 500 can generate conversation that goes beyond 500 tokens meaning that the conversation not only retains information but also adds necessary reasoning resulting in more tokens.

## C.2 CONVERSATION LENGTH VS MODEL PERFORMANCE

As shown in Table 1, LONGEST CONVERSATION achieves the best results among all styles. Since LONGEST CONVERSATION is curated by selecting the longest dialogue (in terms of token count) from seven conversations for a single context, it raises the question of how dialogue length impacts downstream task accuracy.

To explore the relationship between dialogue length and accuracy, we measured the average token length of dialogues across all conversational styles, including LONGEST CONVERSATION. As seen in Table 13, reasoning accuracy does not exhibit a linear correlation with dialogue length. For example, with PROBLEM SOLVING style we can achieve comparable accuracy to LONGEST CONVERSATION even when the average token length for PROBLEM SOLVING is 188 lower than LONGEST CONVERSATION.

| Style | Avg Token Length | Accuracy (Avg-All) |
|---|---|---|
| TWO PROFESSORS | 451.95 | 29.12 |
| TWO STUDENTS | 452.17 | 32.65 |
| PROBLEM SOLVING | 465.29 | 33.38 |
| TEACHER STUDENT | 494.03 | 32.87 |
| INTERVIEW | 497.21 | 32.12 |
| DEBATE | 511.90 | 33.11 |
| LAYMAN KNOWALL | 630.23 | 31.74 |
| LONGEST CONVERSATION | 653.48 | 34.08 |

Table 13: **Conversation Length vs Downstream Task Accuracy:** Conversation length is not correlated with downstream task accuracy.

This highlights that the conversation length is not the only important factor to attain the maximum gain in reasoning ability. As mentioned in Section 5, the structure and dynamics of the conversations also play a crucial role in maximizing reasoning gains.

## C.3 CONVERSATION QUALITY ASSESSMENT

While the conversations generated by the LLM typically appear coherent, there are instances where the conversation fails to preserve the context or lacks grounding to the source material. In some cases, conversations may even be incomplete. Detecting poor-quality generation becomes challenging at scale. To address this, we explore two quality-filtering approaches:

**Heuristic Filtering.** We employ a simple heuristic based on token length. Given that the input context is limited to a maximum of 500 tokens and split into subcontexts of 500 tokens each to maximize information retention, we discard any generated conversations that fall below 50 tokens. This ensures that minimal information loss is detected early.

**LLM-based Scoring.** For a more comprehensive assessment, we use an LLM to score the quality of the generated conversations. We introduce four key metrics for evaluation:

- **Correctness**: Verifies that all information, such as numbers and parameters, is accurately reflected in the conversation.
- **Faithfulness**: Ensures the conversation remains grounded in the context provided.
- **Information Preservation**: Checks whether all relevant facts and knowledge from the original context are retained in the conversation.
- **New Knowledge**: Evaluates whether the conversation introduces additional explanations, reasoning, or definitions not present in the raw input.

Given a raw context and its corresponding conversation, we ask $\mathcal{M}$ to rate the conversation on a scale of 1 to 5 in each of four metrics, with 1 representing poor quality and 5 representing the best possible conversation. To determine the overall quality, we compute the average score across the metrics and choose conversations with average scores more than or equal to 3. Additionally, we utilize the prompt from the FineWebEdu (Penedo et al., 2024) annotation framework to further check the correlation between two scoring approaches. In Figure 5, we plot the scores for 140K conversations using FineWebEdu metrics and our metrics. It is clearly visible from the figure is that LLM tends to rate its own generation higher almost all the time resulting in a skewed distribution of rating. Around 96% of conversations are labelled as high quality. However, compared to FineWebEdu, our metric results in less skewed distribution—making our approach more suitable for evaluating synthetic data derived from a seed corpus.

To further investigate, we choose 20 contexts and their corresponding conversations and manually label them on the above four metrics. We later pass these samples to LLM to obtain the quality scores. The correctness and faithfulness metrics were consistently high, with LLM showing a generation correct 96% of times and human annotators labeling a conversation correct 98% of times (with spearman correlation between two being 0.82) which validates the quality and reliability of the generated synthetic dialogues. When comparing the overall human scores with those from the LLM across the four metrics, we observe a weak correlation between two sets (Spearman's $\rho = 0.03$) and the reasoning behind them.

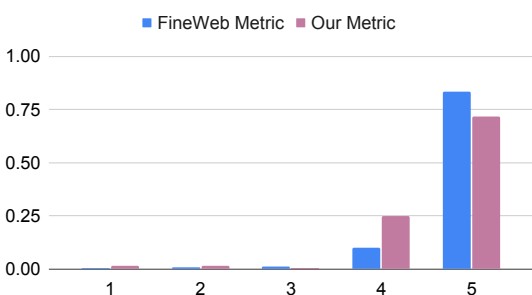

Figure 5: LLM tends to rate its generation higher most of the times.

Human annotators prioritized the information preservation metric, while the LLM often overlooked minor information loss. Additionally, the interpretation of "New Knowledge" differed between humans and the LLM. Humans valued extra reasoning and explanation as forms of new knowledge, whereas the LLM assigned higher "New Knowledge" scores to conversations containing out-of-context information that is difficult to verify. Given these differences in the results from human and LLM-based quality filtering, we use simple heuristic filtering in this study and plan to explore other approaches in the future.

## C.4 COMPARE WITH DEEPSEEKMATH

To asses the quality of our data, we run pre-training experiments to compare MIND-OWM with the recently released DEEPSEEKMATH (Shao et al., 2024). The DEEPSEEKMATH approach is iterative. They construct a dataset for binary classification consisting of 500K positive data points randomly sampled from OpenWebMath (the seed corpus) and 500K negative data points randomly sampled from CommonCrawl. They train a fastText (Joulin, 2016) classifier on these data which they then use to extract samples from CommonCrawl as math content. All CommonCrawl domains for which over

10% of the existing web pages have been extracted are at this point understood to be math-related. URLs which are associated with these domains but which have yet to be collected are manually labeled as math content. The web pages hosted at these addresses are added to the seed corpus and the classifier is retrained. DEEPSEEKMATH performs 4 rounds in total resulting in the DEEPSEEK-MATH Corpus, consisting of some 120B math tokens. They continuously train a partially converged 7B DEEPSEEKCODER-V1.5 model on a 500B token blend to attain the DEEPSEEKMATH model and achieve substantial improvement on several math tasks. In contrast, MIND proposes a simple alternative for generating high-quality math data that boosts the mathematical reasoning ability of LLM given access to a small seed corpus.

As the DEEPSEEKMATH dataset is not public, we replicate our previous blend, $\mathcal{D} = \{\mathcal{X} \cup \mathcal{R}_{pt}\}$, where $\mathcal{X} = \{$MIND-OWM-4B (conversations of all styles except the TWO STUDENTS one) $\cup$ MIND-OWM-14B (TWO STUDENTS conversations)$\}$. We maintain a 2:1 ratio of $\mathcal{X}$ and $\mathcal{R}_{pt}$ in the training blend. Similar to the approach of DEEPSEEKMATH, we take a converged DEEPSEEKCODER-V1.5 model as $\mathcal{C}$ — the unconverged model weights are unpublished as far as we are aware — and convert the model weights to a format compatible with Megatron-LM, which serves as our training framework, before continuously training for 500B tokens. We use a cosine learning rate schedule with a 19B token linear ramp-up, a maximum learning rate of 3e-4, and a minimum learning rate of 3e-6, and we anneal the learning rate over 500B tokens. We use Adam with parameters $\beta_1 = 0.9$ and $\beta_2 = 0.95$, a weight decay of 0.1, a gradient clipping threshold of 1.0, a sequence length of 4096, and a global batch size of 2304 sequences.

| Dataset | Tokens | GSM8K | MATH | MMLU-STEM | MMLU | GENERAL REASONING (Avg) | Avg-Math | Avg-All |
|---|---|---|---|---|---|---|---|---|
| DEEPSEEKMATH (Shao et al., 2024) | | 59.29 | 4.37 | 55.41 | 54.98 | 55.94 | 39.69 | 43.64 |
| MIND-OWM-4B/14B [Combinations*] | 500B | 57.32 | 2.36 | 51.95 | 56.54 | 59.16 | 37.21 | 43.84 |

Table 14: **DEEPSEEKMATH vs All Synthetic Conversations.** A model trained on conversations generated by MIND from a small seed corpus can achieve math accuracy comparable to the DEEPSEEKMATH model trained on 120B unique tokens.

From Table 14, we can see that a model trained on conversations which MIND generated given a small seed corpus can attain math accuracies comparable to the DEEPSEEKMATH model with access to 120B unique math tokens in its continuous training blend. In fact, we outperform DEEPSEEK-MATH in MMLU and general reasoning tasks, reaching higher average accuracy across all tasks. This underscores the quality of MIND generated conversations and signifies the efficacy of MIND in improving mathematical reasoning ability of LLM when the underlying raw data is limited.

In contrast to our prior $\mathcal{C}$, DEEPSEEKMATH-7B LLM is a strong math baseline that has been specifically designed for addressing mathematical reasoning ability and surpasses Azerbayev et al. (2023b), Team (2024a), Jiang et al. (2023), Lewkowycz et al. (2022a), Javaheripi et al. (2023), Dubey et al. (2024) [8B] base models on diverse math tasks. To evaluate the effectiveness of MIND with stronger pretrained model, we perform an additional experiment, similar to our training setup in Section 3.1 using $\mathcal{C}$ = DEEPSEEKMATH-7B. Specifically, we have continuously trained the $\mathcal{C}$ on 500B tokens maintaining a 2:1 ratio of math ($\mathcal{R}$) and 13 CC ($\mathcal{R}_{pt}$) dataset where the total blend is $\mathcal{D} = \{\mathcal{R} \cup \mathcal{R}_{pt}\}$. We conduct two experiments by alternating $\mathcal{R}$ with raw (OWM-14B) and $\mathcal{X}$.

| Dataset | Tokens | GSM8K | MATH | MMLU-STEM | MMLU | GENERAL REASONING (Avg) | Avg-Math | Avg-All |
|---|---|---|---|---|---|---|---|---|
| OWM-14B | | 39.42 | 1.59 | 49.92 | 52.87 | 55.47 | 30.31 | 37.34 |
| MIND-OWM [ALL CONVERSATIONS] | 500B | 57.32 | 2.36 | 51.95 | 56.54 | 59.16 | 37.21 | 43.84 |

Table 15: **Training DEEPSEEKMATH-7B with Raw Data vs All Synthetic Dialogues.** A strong pretrained LLM continuously trained on conversations generated by MIND provides significant boost in math accuracy than the same model trained on raw data—showing the effectiveness of MIND regardless of pretraining model quality.

As shown in Table 15, model trained on MIND-OWM data shows consistent improvement over model trained on raw data—resulting in 17.90% gain on GSM8K, 6.90% average improvement across math

tasks and 3.43% average improvement across ten general reasoning tasks. These results further solidifies the effectiveness of MIND regardless of the quality of the pretrained model.

## C.5 CONVERSATIONS ON CODE TASKS

Unlike raw data, conversations tend to break down the context into sub-context and participants exchange their reasoning about the sub-context in a single turn. This feature is particularly useful for mathematical or logical reasoning which require step-by-step reasoning. However, this structure might hurt performance of LLM in domains where sequence of context needs to be preserved such as in codes. To further investigate the impact of conversational data on the coding capabilities of LLM, we conduct an evaluation of models trained on both raw and synthetic data across four established coding benchmarks: HumanEval (Chen et al., 2021), MBPP (Austin et al., 2021), HumanEval+, and MBPP+ (Liu et al., 2024). These benchmarks are specifically designed to assess the model's ability to generate functional code in response to given prompts.

Our results, as presented in Table 16, demonstrate that conversational synthetic data does not enhance coding performance. This is largely due to the way conversations tend to fragment code, wrapping it in natural language and thereby obscuring the intended sequence and logic inherent in programming tasks. Consequently, while conversations may be effective in contexts that benefit from collaborative reasoning, they are not suited for preserving the integrity of code, leading to diminished performance in coding benchmarks.

| Dataset | Style | HumanEval | HumanEval+ | MBPP (Sanitized) | MBPP+ | Avg-All |
|---|---|---|---|---|---|---|
| OWM-4B | Raw | 12.20 | 10.98 | 23.74 | 0.00 | 11.73 |
| Rephrase-OWM-4B | Rephrase | 5.49 | 4.27 | 20.23 | 0.53 | 7.63 |
| MIND-OWM-4B | TWO PROFESSORS | 8.54 | 4.88 | 20.62 | 0.00 | 8.51 |
| | TEACHER STUDENT | 13.41 | 9.76 | 26.46 | 0.26 | 12.47 |
| | TWO STUDENTS | 10.37 | 7.93 | 26.07 | 0.26 | 11.16 |
| | LAYMAN KNOWALL | 10.37 | 8.54 | 26.46 | 0.79 | 11.54 |
| | DEBATE | 11.59 | 9.15 | 24.90 | 0.26 | 11.48 |
| | INTERVIEW | 7.32 | 4.88 | 23.35 | 0.26 | 8.95 |
| | PROBLEM SOLVING | 9.76 | 9.15 | 24.51 | 0.26 | 10.92 |
| | LONGEST CONVERSATION | 9.15 | 7.32 | 28.40 | 0.53 | 11.35 |
| | ALL CONVERSATIONS | 12.20 | 9.15 | 28.02 | 0.53 | 12.48 |
| OWM-4B+MIND-OWM-4B [1:1] | Combination | 13.41 | 10.98 | 23.35 | 0.00 | 11.94 |
| OWM-4B+MIND-OWM-4B [Concat] | | 10.37 | 7.93 | 31.52 | 0.00 | 12.46 |

Table 16: **Results of 7B LLM on Code Tasks:** Conversations and rephrases are ineffective for code benchmarks.

Interestingly, we also observe that rephrasing, which resembles raw data more closely in structure, further degrades coding accuracy. Our qualitative analysis of the rephrased documents reveals that the conversation generator (LLAMA3-70B-INSTRUCT) often removes critical elements such as important tags and code indentation, which are essential for comprehending and executing code correctly. This finding underscores the necessity of designing code-specific prompts that retain the structural components vital for coding tasks.

## D  DIALOGUE EXAMPLES

**RAW TEXT**

# CAT Quantitative Aptitude Questions | CAT Ratios, Mixtures, Alligations and Averages Questions
###### CAT Questions | Mixtures and Alligations | Mixtures - Range
The question is from CAT Ratio, Mixtures and Averages. Given the details about two alloys, we need to determine the possible values the weight of an ally can take. CAT exam is known to test on basics rather than high funda ideas. A range of CAT questions can be asked from Ratios and Proportions, Mixtures, Alligations and Averages. Make sure you master the topics. 2IIMs CAT questions bank provides you with CAT questions that can help you gear for CAT Exam CAT 2020.
Question 16: 100 kgs of an alloy of tin and lead in the ratio 1:3 is mixed with x kgs of an alloy of tin and lead in the ratio 3:2. If the overall alloy should contain between 40% and 50% tin, what is the range of values x can take?
1. $100 \text{ kgs} \leq x \leq 200 \text{ kgs}$
2. $80 \text{ kgs} \leq x \leq 240 \text{ kgs}$
3. $110 \text{ kgs} \leq x \leq 220 \text{ kgs}$
4. $75 \text{ kgs} \leq x \leq 250 \text{ kgs}$
## Best CAT Coaching in Chennai
#### CAT Coaching in Chennai - CAT 2020Online Batches Available Now!

**SYNTHETIC DIALOGUE [TWO PROFESSORS ]**

**Professor 1:** Good morning, colleague. I'd like to discuss a question from CAT Ratio, Mixtures, and Averages. It's about two alloys of tin and lead, and we need to determine the possible values of the weight of an alloy.
**Professor 2:** Ah, yes. I'm familiar with this type of question. It's a classic example of Mixtures and Alligations. Please go ahead and share the details of the question.
**Professor 1:** We have 100 kgs of an alloy of tin and lead in the ratio 1:3, which means it has 25 kgs of tin and 75 kgs of lead. This alloy is mixed with x kgs of another alloy of tin and lead in the ratio 3:2. The overall alloy should contain between 40% and 50% tin. We need to find the range of values x can take.
**Professor 2:** That's a great problem. We can start by analyzing the second alloy. If it has a ratio of 3:2, that means it has 3x/5 kgs of tin and 2x/5 kgs of lead, with a total weight of x kgs.
**Professor 1:** Exactly. Now, when we mix these two alloys, how does the mixture change? We need to find the overall percentage of tin in the resulting alloy.
**Professor 2:** Right. Let's think about it. If the overall alloy should contain between 40% and 50% tin, what does that imply about the range of values x can take?
**Professor 1:** That's the key question. If we can determine the range of x, we can answer the question. Do you think it's possible to narrow down the options to one of the given choices?
**Professor 2:** Yes, I think so. Let's analyze the options. We have 1. $100 \text{ kgs} \leq x \leq 200 \text{ kgs}$, 2. $80 \text{ kgs} \leq x \leq 240 \text{ kgs}$, 3. $110 \text{ kgs} \leq x \leq 220 \text{ kgs}$, and 4. $75 \text{ kgs} \leq x \leq 250 \text{ kgs}$. Which one do you think is the correct answer?
**Professor 1:** Well, after analyzing the problem, I think the correct answer is option 2: $80 \text{ kgs} \leq x \leq 240 \text{ kgs}$. What do you think?
**Professor 2:** I agree with you. The correct answer is indeed option 2. Well done! This question is a great example of how CAT exam tests our understanding of basics rather than high-funda ideas.

## RAW TEXT

?
Free Version
Moderate
# Interpretation of Equilibrium from Graph of Concentration vs Time
CHEM-NWU9LE
Consider the graph below of a chemical reaction observing the concentration of two chemical species with time.
Which species is the reactant? $$
Which species is the product?
$$
At what region does the solution appear to be in equilibrium? $$
What is the $K_c$ for the reaction?

## SYNTHETIC DIALOGUE [TEACHER STUDENT ]

**Student:** Hi, teacher. I'm having trouble understanding this graph of a chemical reaction. Can you help me?
**Teacher:** Of course! What's your first question about the graph?
**Student:** Okay, so which species is the reactant?
**Teacher:** Let's take a look at the graph. The species whose concentration is decreasing with time is the reactant. Can you identify which one that is?
**Student:** Ah, I think it's... (looks at the graph) ...the one whose concentration starts high and decreases.
**Teacher:** That's correct! The reactant is the species whose concentration starts high and decreases with time. Now, what's your next question?
**Student:** Which species is the product?
**Teacher:** Good question! The product is the species whose concentration starts low and increases with time. Can you identify which one that is?
**Student:** Yeah, I think it's... (looks at the graph) ...the one whose concentration starts low and increases.
**Teacher:** That's right! The product is the species whose concentration starts low and increases with time. Now, what's your next question?
**Student:** At what region does the solution appear to be in equilibrium?
**Teacher:** Look at the graph again. At what point does the concentration of both species seem to be constant, not changing with time?
**Student:** Hmm... I think it's... (looks at the graph) ...after a certain time, when the lines are flat.
**Teacher:** Exactly! The solution appears to be in equilibrium when the concentration of both species is constant, which is after a certain time when the lines are flat. Now, what's your next question?
**Student:** What is the Kc for the reaction?
**Teacher:** Unfortunately, the graph doesn't provide enough information to calculate the Kc value. We would need more data or information about the reaction to determine the equilibrium constant, Kc.
**Student:** Okay, I think I understand now. Thank you for your help!

### RAW TEXT

What's the probability that: The person is not age 18 to 34 and does not own an iPhone? The presidential election is coming.

Five survey companies (A, B, C, D, and E) are doing survey to forecast whether or not the Republican candidate will win the election.

Each company randomly selects a sample size between 1000 and 1500 people.

All of these five companies interview people over the phone during Tuesday and Wednesday. The interviewee will be asked if he or she is 18 years old or above and U.S. citizen who are registered to vote. If yes, the interviewee will be further asked: will you vote for the Republican candidate?

On Thursday morning, these five companies announce their survey sample and results at the same time on the newspapers.

The results show that a% (from A), b% (from B), c% (from C), d% (from D), and e% (from E) will support the Republican candidate.

The margin of error is plus/minus 3% for all results.

Suppose that $c>a>d>e>b$. When you see these results from the newspapers, can you exactly identify which result(s) is (are) not reliable and not accurate?

That is, can you identify which estimation interval(s) does (do) not include the true population proportion?

If you can, explain why you can, if no, explain why you cannot and what information you need to identify.

Discuss and explain your reasons. You must provide your statistical analysis and reasons.

### SYNTHETIC DIALOGUE [TWO STUDENTS ]

**Turn 1**
Alex: Hey Ben, have you started working on our assignment about the presidential election survey?
Ben: Yeah, I've been going over the context. It seems like we need to identify which survey result is not reliable and accurate.
Alex: Exactly! Let's break it down. We have five survey companies, A, B, C, D, and E, each with a sample size between 1000 and 1500 people.
Ben: Right, and they all have a margin of error of plus/minus 3%.
**Turn 2**
Alex: Okay, so we're given that c > a > d > e > b. What does this tell us?
Ben: It tells us the order of the proportions of people who will support the Republican candidate according to each survey company.
Alex: That's right. And since the margin of error is the same for all companies, we can assume that the true population proportion should lie within a similar range for all companies.
**Turn 3**
Ben: But wait, if the true population proportion lies within a similar range, then shouldn't all the estimation intervals overlap?
Alex: Exactly! If the estimation intervals don't overlap, it means that at least one of them is not reliable and accurate.
Ben: So, we need to check if the intervals overlap. How do we do that?
**Turn 4**
Alex: Let's calculate the estimation intervals for each company. For company A, it's a% ± 3%, and similarly for the others.
Ben: Okay, got it. So, we have a% ± 3%, b% ± 3%, c% ± 3%, d% ± 3%, and e% ± 3%.
Alex: Now, let's check if these intervals overlap. Since c > a > d > e > b, the interval for company C should be the highest, and the interval for company B should be the lowest.
**Turn 5**
Ben: I see what you're getting at. If the intervals overlap, then the highest value of the lowest interval (company B) should be greater than or equal to the lowest value of the highest interval (company C).

Alex: That's right! If that's not the case, then we know that at least one of the intervals is not reliable and accurate.

Ben: Let's plug in some numbers and see if we can identify which one is not reliable.

**Turn 6**

Alex: Alright, let's say the results are a = 45%, b = 38%, c = 50%, d = 42%, and e = 40%. Can we identify which one is not reliable?

Ben: Let's calculate the estimation intervals... (calculates)...and check if they overlap.

Alex: (checks the calculations) Ah, look at that! The interval for company B ($38\% \pm 3\%$) doesn't overlap with the interval for company C ($50\% \pm 3\%$).

Ben: That means company B's result is not reliable and accurate!

**Turn 7**

Ben: But why can't we identify the other unreliable results, if any?

Alex: That's because we only have the relative order of the proportions, not the exact values. We need more information to identify other unreliable results, if any.

Ben: I see. So, we can only identify company B's result as not reliable and accurate, but we can't say for sure about the others.

Alex: That's correct. We've done our best with the given information.

RAW TEXT

# Decimal to Percent Calculator.
The calculator will convert decimal into percent, with steps shown.
If the calculator did not compute something or you have identified an error, or you have a suggestion/feedback, please write it in the comments below.
Convert 1.4 into decimal. ## Solution From the converting decimal to percent note, we know that to obtain a percent, we need to multiply a decimal by 100.
But multiplying by 100is equivalent to moving a decimal point 2 positions to the right.

SYNTHETIC DIALOGUE [LAYMAN KNOWALL ]

**Turn 1**
Layman: Hi, I'm not sure what this calculator does. Can you explain it to me?
Me: This calculator converts decimals into percents. It shows the steps to do the conversion.
**Turn 2**
Layman: Okay, got it. So what's the problem we're trying to solve?
Me: We're trying to convert 1.4 into a percent.
**Turn 3**
Layman: How do we do that? Is there a formula or something?
Me: Yes, there is. According to the note on converting decimal to percent, we need to multiply the decimal by 100 to obtain a percent.
**Turn 4**
Layman: Why do we need to multiply by 100? What's the logic behind it?
Me: Well, I'm not sure about the logic, but the note says that's what we need to do. But I can tell you that multiplying by 100 is equivalent to moving the decimal point 2 positions to the right.
**Turn 5**
Layman: What do you mean by moving the decimal point 2 positions to the right? Can you explain that?
Me: Sure. When you multiply a number by 100, the decimal point shifts 2 places to the right. For example, if you have 1.4 and you multiply it by 100, the decimal point will move 2 places to the right, making it 140.
**Turn 6**
Layman: Ah, I think I understand now. So, if we multiply 1.4 by 100, what do we get?
Me: That's right! If we multiply 1.4 by 100, we get 140. And that's the equivalent percent value of the decimal 1.4.

RAW TEXT

Try to finish the derivation and see how it applies to the sum you're trying to compute. Sorry if this is hard to understand.
7. Jun 14, 2009
### jgens
Cyosis, why would he/she use the sum for the first n natural numbers when his sum asks for odd natural numbers only?
8. Jun 14, 2009
### Cyosis
If you know the sum for the first n natural numbers you can quickly solve the sum for the first odd numbers. Had he known it, it would have been the easiest way to calculate the sum he's interested in.
9. Jun 14, 2009
### Manchot
The story is that a six-year-old Gauss was told by his teacher to add the numbers 1 to 100 as a form of busywork. In seconds, he came up with the answer, much to his teacher's surprise. What he did was notice that since 1+100=101, 2+99=101, 3+98=101, etc., the sum becomes a simple multiplication: there are 50 of the above p̈airs,s̈o the sum is just 50*101 = 5050. Anyway, that's the intuition behind the identity that Cyosis posted. You can pretty much use the exact same intuition.
10. Jun 14, 2009
### gabrielh
Okay thanks, I get it now.
11. Jun 14, 2009

SYNTHETIC DIALOGUE [DEBATE ]

**jgens**: I don't understand why Cyosis suggested using the sum of the first n natural numbers to solve the problem. The original problem asks for the sum of the first odd natural numbers, not all natural numbers.

**Cyosis**: I see what jgens is saying, but I think my approach is still valid. If you know the sum of the first n natural numbers, you can easily derive the sum of the first odd natural numbers. It's actually the easiest way to calculate the sum.

**Manchot**: I think I can help clarify things. The story about Gauss adding numbers from 1 to 100 is relevant here. He noticed that 1+100=101, 2+99=101, and so on, which makes the sum a simple multiplication problem. There are 50 such pairs, so the sum is 50*101 = 5050. This is the intuition behind Cyosis' suggestion.

**jgens**L I see what Manchot is saying, but I'm still not convinced. How does this story about Gauss relate to the sum of odd natural numbers?

**Cyosis**: Well, jgens, the same intuition applies here. We can use a similar pairing method to calculate the sum of odd natural numbers. It's not a direct application, but the principle is the same.

**gabrielh**: I think I'm starting to get it now. So, we can use the sum of the first n natural numbers as a stepping stone to calculate the sum of the first odd natural numbers, and the Gauss story provides the intuition behind this approach.

**Manchot**: Exactly, gabrielh! That's the idea. By understanding the sum of the first n natural numbers, we can derive the sum of the first odd natural numbers using a similar method.

**Cyosis**: I'm glad we could clarify things. The key is to recognize that the sum of the first n natural numbers is a more general problem that can be used to solve the more specific problem of the sum of the first odd natural numbers.

RAW TEXT

The Cryptographic Lens
Shafi Goldwasser - Colloquium Lecture - POSTPONED!
Tuesday, 21.10.2014, 14:30
Room 337-8 Taub Bld.

...
ceClub: A Centralized Žero-QueueŇetwork Architecture
Jonathan Perry (MIT, CSAIL)
Monday, 06.10.2014, 11:00
EE Meyer Building 861
Current datacenter networks inherit the principles that went into the design of the Internet, where packet transmission and path selection decisions are distributed among the endpoints and routers. Instead, we propose that each sender should delegate control—to a centralized arbiter—of when each packet should be transmitted and what path it should follow. Fastpass is a datacenter network architecture built using this principle. Fastpass incorporates two fast algorit...
Parameterized Automata Constructions and Their Applications
Ran Ben-Basat
Monday, 22.09.2014, 15:30
Taub 701
Parameterization is a useful tool for handling NP-hard problems in the real world. It aims to reduce the running times of algorithms for such problems, by confining the combinatorial explosion to some parameter k. As this parameter is often significantly smaller than the input size, it allows to develop practical algorithms for non-trivial classes of instances for these problems. In this talk we present a novel framework for developing parameterized algorithms, using construction...
Observing the Observers: Social Context Analysis Using Computer Vision
Meir Cohen
Wednesday, 10.09.2014, 12:00
Taub 601
It is quite common that multiple human observers attend to a single point of interest.
Mutual awareness activity (MAWA) refers to the dynamic of this social phenomena.
A peak of a MAWA is known as a mutual awareness event (MAWE) and can be interpreted as a b̈uzzëvent, which draws the attention of many observers.
A preferred way to monitor those social phenomenon is with a camera that captures the human observers while they observe the activity in the scene.
Our work studies the ...

SYNTHETIC DIALOGUE [INTERVIEW ]

**Subject Matter Expert:** Unfortunately, the lecture was postponed, so we don't have any information about the specific topic she was going to cover. However, the title of the lecture was T̈he Cryptographic Lens,ẅhich suggests that it might have been related to cryptography and its applications.
**Interviewer:** That's a shame it was postponed. Moving on, let's talk about the ceClub lecture by Jonathan Perry from MIT, CSAIL. Can you summarize the main idea behind his proposed datacenter network architecture, Fastpass?
**Subject Matter Expert:** Yes, certainly. Jonathan Perry proposed a centralized žero-queueňetwork architecture called Fastpass, which is a departure from the traditional distributed approach used in Internet design. In Fastpass, each sender delegates control to a centralized arbiter, which decides when each packet should be transmitted and what path it should follow. This approach aims to improve the efficiency of datacenter networks.
**Interviewer:** That's fascinating. What can you tell me about the talk by Ran Ben-Basat on parameterized automata constructions and their applications?

**Subject Matter Expert:** Ran Ben-Basat discussed the concept of parameterization, which is a useful tool for handling NP-hard problems in the real world. The idea is to reduce the running times of algorithms for these problems by confining the combinatorial explosion to some parameter k. He presented a novel framework for developing parameterized algorithms using construction-based methods.

**Interviewer:** I see. Now, let's switch gears and talk about the lecture by Meir Cohen on social context analysis using computer vision. Can you explain what mutual awareness activity (MAWA) and mutual awareness event (MAWE) refer to in this context?

**Subject Matter Expert:** Yes. Mutual awareness activity (MAWA) refers to the dynamic of multiple human observers attending to a single point of interest. A peak of this activity is known as a mutual awareness event (MAWE), which can be interpreted as a buzzëvent that draws the attention of many observers. Meir Cohen's work focuses on monitoring and analyzing these social phenomena using computer vision, specifically by capturing the human observers with a camera while they observe the activity in the scene.

RAW TEXT

Note: At section X = 12 m, the speed is close to 1.5 m / s Thus for a return trip of 2x20m = 40 m to return to the profile X = 14m, the wave takes 27 s. Thus, the graph of Figure. 11 is located before the (eventual) reflected wave coming from the absorbing upstream boundary, which can disturb the sensors.
## Comparative analysis of records
We have exploited some of the tests in a transversal manner, to identify particular behaviors of the surface waves:
• impact of the variation of the amplitude for several tests characterized by the same parameters but with roughnesses, or geometries either uniform, or linear
• impact of the change in the period
• impact of the variation of the initial amplitude
• comparison for the same test carried out in uniform or converging section
• impact of roughness variation
### Impact of the variation of the amplitude for several tests with the same parameters
We have compared several tests characterized by D70, by Ab absorbing boundary conditions and for the same period T5. These are tests D70A5T5, D70A2T2 and D70A5T7 in RoAbUn, SmAbCo and SmAbUn conditions
Figure 16 Figure 17 Figure 18
• Figure 16: amplitudes variations for the 3 runs are increasing according to the slope, which is in accordance with the theory.
• Figure 17: the initial amplitudes of the waves generated by the wave maker are not always well respected: all the curves of the same diagram should start from the same point to X = 0
• Figure 18: there is a sharp change in amplitude at the arrival at the top of the slope.

SYNTHETIC DIALOGUE [PROBLEM SOLVING ]

**User:** Hi, I'm analyzing some data on surface waves and I'd like to discuss some observations with you. Can you help me understand the impact of varying amplitude on surface waves?
**AI Assistant:** I'd be happy to help. From the context, I see that you have several tests with the same parameters but different amplitudes. Specifically, you have tests D70A5T5, D70A2T2, and D70A5T7 in RoAbUn, SmAbCo, and SmAbUn conditions. What would you like to know about these tests?
**User:** Yes, that's correct. I'm looking at Figures 16, 17, and 18. Can you explain what's happening in Figure 16?
**AI Assistant:** In Figure 16, the amplitudes of the waves are increasing according to the slope, which is consistent with theoretical expectations. This suggests that the amplitude of the wave is directly related to the slope of the surface.
**User:** That makes sense. But what about Figure 17? The initial amplitudes of the waves generated by the wave maker don't seem to be consistent.
**AI Assistant:** That's a good observation. Yes, the initial amplitudes of the waves don't start from the same point at X = 0, which is unexpected. This could be an issue with the wave maker or the experimental setup.
**User:** Okay, got it. And what about Figure 18? There seems to be a sudden change in amplitude at the top of the slope.
**AI Assistant:** Yes, that's correct. Figure 18 shows a sharp change in amplitude at the arrival at the top of the slope. This could be due to the interaction between the wave and the slope, causing a sudden increase or decrease in amplitude.
**User:** That's helpful. So, what can we conclude about the impact of varying amplitude on surface waves?
**AI Assistant:** Based on these figures, it seems that the amplitude of the wave is directly related to the slope of the surface, and changes in amplitude can affect the behavior of the wave. However, we also need to consider the potential issues with the wave maker or experimental setup that may be affecting the results.
**User:** Okay, that's a good summary. Thanks for your help in analyzing these figures!

