# OpenReview forum: "MIND: Math Informed syNthetic Dialogues for Pretraining LLMs"
_ICLR.cc/2025/Conference — ICLR 2025 Poster_

### Official Review · Reviewer_pFmJ · 2024-10-30

**Soundness:** 3
**Presentation:** 3
**Contribution:** 2
**Rating:** 5
**Confidence:** 4

**Summary:**

This paper introduces a new method of generating synthetic math data for continued pretraining. The authors design a conversation setting to let two speakers talk about a given text in the math domain. Each speaker can be a teacher, student, professor, etc. Seven types of person pairs are used as prompts (e.g., teacher-student), though it seems that professor-professor is less helpful from the perspective of downstream performance.

**Strengths:**

Experiments on two pretraining corpora demonstrate the helpfulness of the proposed method on reasoning tasks.

This method can also be used to clean pre-training corpora.

**Weaknesses:**

I have several concerns/questions regarding the motivation/details.

- The quality of the synthetic math data: The authors introduced that heuristics are applied to filter the generated conversations and provided the similarity between raw text and the synthetic dialogues. I'm wondering about the length difference between the original text and the corresponding conversation (on average), and it is unclear the degree of hallucination issues in the generated text.

- The impact of knowledge distillation: LLaMa3-70B-Instruct is prompted with different persona-pair for conversation generation. However, its contribution is not well justified. The authors may consider weaker models such as LLaMa3-8B or even an SFT one based on the pre-trained model in this paper.

- A stronger baseline may be considered: The authors only compare their method with a rephrasing text baseline. The conversation format explored in this paper seems to be useful for improving the multi-turn instruction following ability. A more advanced baseline can be "given a text, (follow-up) questions and their answers are generated automatically" as other continued pretraining studies (e.g., [1])

- Please elaborate on the details in the evaluation:
    * e.g., specify zero-shot or few-shot for different tasks. I'm wondering whether the gains will be decreased in the few-shot setting.
    * The longest-conversation achieves the best performance (Table 1). Why is "student-student" considered for experiments on 14B and ablation studies?

- The authors may consider more diverse persona combinations for diversity and data scale-up.

**Questions:**

please refer to the previous section.

---

### Official Review · Reviewer_ikXv · 2024-11-04

**Soundness:** 2
**Presentation:** 3
**Contribution:** 2
**Rating:** 5
**Confidence:** 4

**Summary:**

This paper proposes MIND to generate large-scale, high-quality synthetic math dialogues to pretrain LMs and enhance their mathematical reasoning abilities. MIND prompts a pretrained LLM (llama3-70B-instruct) to convert raw mathematical text into structured multi-turn conversations using diverse conversational styles. These synthetic dialogues break down complex problems step-by-step while injecting complementary explanations and reasoning. The authors generated 64B tokens of synthetic data using the 14B token OpenWebMath corpus. They conducted experiments with varying conversational styles and participants to assess impact on reasoning. Models pretrained on MIND-generated dialogues outperformed those trained on raw data in mathematical reasoning and general reasoning. In summary, MIND provides a way to upsample limited high-quality math data into structured dialogues that embed multi-hop reasoning.

**Strengths:**

Originality: The paper presents an approach (MIND) to generate high-quality synthetic math dialogues that improve math reasoning in LMs. Compared to prior work on synthetic pretraining data that mostly rephrases raw text, MIND adds semantic variations and step-by-step reasoning that are crucial for complex math problem solving.

Quality: The paper is thorough in evaluating MIND across multiple dimensions - testing different conversational styles, scaling behavior, and applicability to varying seed corpora. Ablations provide good insights, like the importance of knowledge gaps between dialogue participants.

Clarity: The paper is easy to read. Key aspects of the approach, experiments and results are clearly explained.

Significance: This work demonstrates the potential of structured conversational data to enhance reasoning in language models, especially when domain-specific high-quality data is limited.

**Weaknesses:**

- Since the method uses the LLAMA3-70B-INSTRUCT model to generate conversations, it is unclear whether the improvements in downstream reasoning tasks come from the quality of the generated dialogues or are simply a result of model distillation from the powerful LLAMA3-70B model. The authors should investigate this and isolate the impact of the MIND-generated dialogues from the influence of the underlying LLM.

- The experiments are based on a single in-house pretrained model checkpoint. It is possible that this model is not very well pretrained, making the improvements from synthetic dialogues appear more significant than they would be for a highly-optimized model. To demonstrate the generality of the MIND approach, the authors can experiment with multiple popular and high-quality pretrained models such as LLAMA-3, Mistral, GEMMA, etc. Consistent improvements across a range of strong baseline models would provide more convincing evidence for the effectiveness of the proposed method.

- The paper focuses on math reasoning. It would be nice to see if MIND generalizes to other technical domains needing step-by-step reasoning, like physics, engineering, coding etc. Some preliminary experiments could help to see the broader applicability.

**Questions:**

- What is the impact of conversation length on model performance? Is there an optimal length?

- How well does this approach generalize to other technical domains beyond mathematics?

- The continued pretraining setup upsamples OpenWebMath but keeps CommonCrawl data constant. How much does the relative ratio of math vs general data impact results?

- Will the data be released?

---

### Official Review · Reviewer_vrdW · 2024-11-04

**Soundness:** 4
**Presentation:** 4
**Contribution:** 3
**Rating:** 8
**Confidence:** 4

**Summary:**

The paper introduces MIND (Math Informed syNthetic Dialogue), a novel method for generating large-scale synthetic dialogues to improve the mathematical reasoning abilities of large language models (LLMs). Traditional synthetic data often fails to enhance complex, multihop reasoning tasks because it doesn't add new knowledge to existing corpora. MIND addresses this by generating structured conversations based on the OpenWebMath (OWM) corpus, resulting in a new dataset called MIND-OWM.

These synthetic dialogues decompose complex math problems into multi-turn conversations, promoting step-by-step reasoning. Experiments reveal that incorporating knowledge gaps between dialogue participants is crucial for generating high-quality math data. Models pretrained on MIND-OWM show significant improvements in mathematical reasoning benchmarks (e.g., GSM8K: +13.42%, MATH: +2.30%), specialized knowledge tasks (MMLU: +4.55%, MMLU-STEM: +4.28%), and general reasoning tasks (+2.51%) compared to models pretrained on raw data alone

**Strengths:**

1. This paper articulates the contributions, methodologies, and results in a clear way. It is pleasant to read.
2. Despite simple and straightforward, strong experimental results showcase the effectiveness of the proposed MIND framework and the data synthesized by it. The results are evaluated on three different math corpus and they all demonstrate substantial improvements by training on the generated data.

**Weaknesses:**

One thing I would like to point out is there is a very relevant work to discuss and compare to.
Dialog Inpainting: Turning Documents into Dialogs (https://proceedings.mlr.press/v162/dai22a.html)
This paper also proposes ways to synthesize conversations from knowledge sources. Although it is not specifically for math, the authors should at least discuss the differences in methods in rebuttals.

**Questions:**

N/A

---

### Comment · Area_Chair_YhkZ · 2024-11-25
**Action Required: Respond to Author Rebuttals - Nov 27**

Dear ICLR Reviewers,

The author discussion phase is ending soon. Please promptly review and respond to author rebuttals for your assigned papers. Your engagement is critical for the decision-making process.

Deadlines:
- November 26: Last day for reviewers to ask questions to authors.
- November 27: Last day for authors to respond to reviewers.
- November 28 - December 10: Reviewer and area chair discussion phase.

Thank you for your timely attention to this matter.

---

### Meta-Review · Area_Chair_YhkZ · 2024-12-21

**Metareview:**

The paper presents MIND (Math Informed syNthetic Dialogue), a novel method for generating large-scale synthetic dialogues to enhance mathematical reasoning capabilities in LLMs. The authors introduce a new math corpus, MIND-OWM, by leveraging the proposed method to generate synthetic conversations based on OpenWebMath. The approach converts raw mathematical text into structured multi-turn conversations using diverse conversational styles and participant knowledge gaps. Some reviewers acknowledge the work's strengths, including clear presentation, comprehensive experimental evaluation across multiple benchmarks, and performance improvements on mathematical reasoning tasks. However, reviewers raise several concerns. First, there are questions about whether the improvements stem from the quality of generated dialogues or are simply a result of knowledge distillation from the powerful LLAMA3-70B model used for generation. The authors could expand the evaluation to newer benchmarks beyond commonly used ones like GSM8K/MMLU/MATH to better demonstrate the method's generalizability. Also, the evaluation details and baselines could be strengthened, particularly regarding zero-shot vs. few-shot performance and comparison with more advanced question-answer generation approaches. Given the authors have addressed some concerns through additional clarifications and analyses, I recommend marginal acceptance to ICLR.

**Additional Comments On Reviewer Discussion:**

see above

---

### Decision · Program_Chairs · 2025-01-22

Accept (Poster)